# Imaging in Hip Arthroplasty Management Part 2: Postoperative Diagnostic Imaging Strategy

**DOI:** 10.3390/jcm11154416

**Published:** 2022-07-29

**Authors:** Charles Lombard, Pierre Gillet, Edouard Germain, Fatma Boubaker, Alain Blum, Pedro Augusto Gondim Teixeira, Romain Gillet

**Affiliations:** 1Guilloz Imaging Department, Central Hospital, University Hospital Center of Nancy, 29 Avenue du Maréchal de Lattre de Tassigny, F-54000 Nancy, France; c.lombard@chru-nancy.fr (C.L.); e.germain@chru-nancy.fr (E.G.); f.boubaker@chru-nancy.fr (F.B.); alain.blum@gmail.com (A.B.); ped_gt@hotmail.com (P.A.G.T.); 2Université de Lorraine, CNRS, IMoPA, F-54000 Nancy, France; pierre.gillet@univ-lorraine.fr; 3Laboratoire de Pharmacologie-Toxicologie, Pharmacovigilance & CEIPA, Bâtiment de Biologie Médicale et de Biopathologie, CHRU de Nancy-Brabois, 5 Rue du Morvan, F-54500 Vandœuvre-Lès-Nancy, France

**Keywords:** hip, arthroplasty, CT, MRI, loosening

## Abstract

Hip arthroplasty (HA) is a frequently used procedure with high success rates, but 7% to 27% of the patients complain of persistent postsurgical pain 1 to 4 years post-operation. HA complications depend on the post-operative delay, the type of material used, the patient’s characteristics, and the surgical approach. Radiographs are still the first imaging modality used for routine follow-up, in asymptomatic and painful cases. CT and MRI used to suffer from metallic artifacts but are nowadays central in HA complications diagnosis, both having their advantages and drawbacks. Additionally, there is no consensus on the optimal imaging workup for HA complication diagnosis, which may have an impact on patient management. After a brief reminder about the different types of prostheses, this article reviews their normal and pathologic appearance, according to each imaging modality, keeping in mind that few abnormalities might be present, not anyone requiring treatment, depending on the clinical scenario. A diagnostic imaging workup is also discussed, to aid the therapist in his imaging studies prescription and the radiologist in their practical aspects.

## 1. Introduction

Hip arthroplasty (HA) is a frequently used procedure with high success rates, its main indications being osteoarthritis, acute fracture, osteonecrosis of the femoral head, and hip dysplasia. Despite its good clinical outcomes, 7% to 27% of patients complain of persistent postsurgical pain 1 to 4 years post-operation, with approximately 12% of the patients describing it as a significant issue [1,2,3,4]. Follow-up surgery is necessary for about 1.3% of all total hip arthroplasty (THA) cases per year, including all types of complications [2]. HA complications depend on the post-operative delay, the type of material used, the surgical approach, and the patient’s comorbidities and activities [5,6]. Imaging in THA patients has several purposes: normal follow-up (to detect complications without clinical symptoms), patients’ complaints (to determine or help to determine the cause of the patients’ symptoms), biopsy or fluid collection guidance, and planning for the treatment of complications (i.e., implants positioning, bone stock evaluation). Therefore, clinicians must be aware of these factors to comprehensively evaluate patients with HA [7] and order prompt imaging workup for an optimal diagnosis. Radiographs are the first imaging modality used for routine follow-up in asymptomatic and painful cases [7]. CT and MRI used to suffer from metallic artifacts, but are nowadays central in HA complications diagnosis [8,9,10,11,12,13,14,15,16,17,18,19,20,21,22,23,24,25,26,27,28,29,30,31,32,33,34,35,36,37,38,39]. Nuclear medicine studies are helpful when CT and MRI are inconclusive when a complication is suspected [40] but are beyond the scope of this paper. Ultrasonography is often limited to evaluating periprosthetic soft tissues and image-guided procedures, especially joint fluid aspiration [13]. All imaging modalities should be considered complementary in terms of a diagnostic strategy. In this review, after a brief reminder about the different types of HA implants, we will focus on HA post-operative normal appearance and complications, via an imaging approach, according to each imaging modality. Each imaging technique’s place and technical aspect will also be discussed.

## 2. Types of HA

### 2.1. Type of Prosthesis and Implants

Acknowledging the prosthesis type is an essential prerequisite to its imaging analysis, concerning its anticipated appearance and potential complications. If not present in the medical file, this information can be reached by a systematic study of the implant’s appearance.

A hip replacement can be either a hemi-prosthesis or a total prosthesis (THA), depending on which part of the joint is replaced [37], with different designs of implants, components, bearing surfaces, fixations, and surgical approaches (posterior, direct-lateral, anterolateral, and anterior [8]). 

THA can be conventional or resurfacing types and replaces both the femoral and acetabular sides of the hip. The component’s analysis must include the acetabular cup, the femoral head, and the femoral stem. Acetabular and femoral components may themselves be modular (neck-head and/or neck-stem junctions) or nonmodular (one piece), the latter being currently uncommon [13]. Therefore, the femoral component might be composed of one, two, or three parts (stem, collar, head) and the acetabular component of one or two (metal-back and insert) pieces. Dual mobility works by containing a three-component system: a socket, a free polyethylene (PE) liner, and a head, allowing mobility between the femoral head and the liner and between the liner and the acetabular cup [41]. “Sandwich” liner refers to a PE insert outlined in its inner face by a ceramic or metallic layer to combine PE elasticity and metal or ceramic resistance, and in its outer surface by an acetabular metal back so that the PE finds itself in a “sandwich” position between two “hard” layers [42]. A resurfacing prosthesis only replaces the proximal femoral head, leaving the native femoral neck and a part of the femoral head intact. This kind of prosthesis is mostly used in younger patients as it preserves bone stock and allows for easier revision than conventional THA [14,32] (Figure 1).

A hemi-prosthesis only replaces the femoral head and can be unipolar (femoral component articulating with the native acetabulum) or bipolar (femoral component articulating with a non-fixated acetabular cup) (Figure 2). Distinguishing a hemi-prosthesis from THA is crucial and can be completed by analyzing acetabular native cartilage and subchondral plate, both preserved in hemiarthroplasty. One must keep in mind that the acetabular cartilage might wear down, leading to a protrusion of the acetabular modular piece, which might be confused with an acetabular THA component. Additionally, a bipolar femoral head has a slightly greater than hemispheric shape, lacks screw holes, and has a smooth outer surface rather than a textured one in case of THA [37]. 

According to Gulow et al., THA femoral components can be classified as: (1) resurfacing endoprostheses anchoring on the epiphysis, (2) collum endoprostheses solely anchoring on the metaphysis, (3) short collum preserving stems anchoring on the metaphysis with short anchorage on the diaphysis, and (4) conventional stems anchoring on the metaphysis with a long diaphyseal anchorage [43].

Conventional stems can be associated with peri-prosthetic fractures, thigh pain, and proximal stress shielding and expose to the loss of bone stock in case of revision surgery. Therefore, short stems (length inferior to 120 mm) have been developed, as they are thought to induce less stress shielding, preserve bone stock, restrict proximo-distal mismatch, and reduce pain. Even though no long-term data are available, clinical and radiological outcomes are promising [44].

### 2.2. Type of Fixation

HA components can be fixed with or without cement, the latter being predominant. Hybrid THA refers to a cemented femoral stem and a cementless acetabular piece, and reverse hybrid THA to the contrary [45]. 

In cemented femoral components, intramedullary plugs are commonly used and are available in a wide variety of materials and shapes, both biodegradable and non-resorbable. Osteolytic changes around biodegradable models have even been described [46]. Some types of restrictors have a metallic marker that can appear at the tip of the stem (Figure 1). Additionally, a proximal and/or distal centralizer can be used to obtain a uniform cement mantle and neutral alignment of the femoral stem [47]. It is composed of polymethylmethacrylate [48] and can appear as a radiolucent zone not to be mistaken for osteolysis (Figure 3). 

In cementless fixation, components stimulate osseous incorporation by their geometry or surface texture and coating. They are impacted for primary fixation, then screws can be added for additional security. 

It has recently been shown that stem design and cementation impacted post-operative femoral ante-torsion, the cemented ones showing the less variability and lowest rates of retro-torsion [49]. On the other hand, acetabular fixation failure was not encountered in cementless implants in Coden’s study, revision being essentially secondary to dislocation [50]. In case of femoral neck fracture in elderly patients, cemented prostheses have better functional outcomes (higher Harris Hip score) and lower rates of revision, fracture, and dislocation than uncemented implants, even though no difference exists in terms of visual analog pain scale, loosening rate, and heterotopic ossification [51]. 

### 2.3. Bearing Surfaces

Multiple combinations are possible depending on the joint side. Polyethylene (PE), metal, and ceramic might be used on the acetabular side, whereas, on the femoral side, only ceramic and metal can be. Combinations including PE and ceramic or metal are called “hard-on-soft” (most commonly used) and those without PE “hard-on-hard”. The bearing surface is a significant factor in determining the longevity of THA, as worldwide concerns regarding expanding burden for revision procedures are emerging, secondary to loosening and osteolysis, depending on the type of bearing couples [52].

The four couples used are metal–PE (MoP), ceramic–PE (CoP), metal–metal (MoM), and ceramic–ceramic (CoC) (Figure 4). However, the optimal bearing surfaces are still under debate. PE liners are often used on the acetabular side, regardless of the artificial head [52]. MoM THA with large femoral head diameters have led to a high rate of adverse reactions to metal debris and pseudo-tumors, leading to an almost worldwide cessation of their use and reinforced surveillance [53]. MoP bearing was known to lead to particles disease secondary to contact pressure-induced wear, but recent data showed that the use of an acetabular ultra-high molecular weight PE cup in association with a titanium alloy femoral head could reduce the wear rate compared to cobalt chromium molybdenum and stainless steel so that it can extend the life of THA, in developing countries, especially for Indonesian and, more widely, Asian people [6]. Additionally, to decrease the amount of PE wear debris, alumina ceramics can be used with newer PE, with good long-term functional and radiologic outcomes, so that CoP is thought to be an excellent bearing couple and accounts for more than half of THA in the USA [52]. CoC bearings have been widely used in Korea for economic purposes but can lead to ceramic fractures and squeaking. Those specific complications rate might be dramatically decreased by the use of CoP bearing surfaces [52]. 

## 3. Imaging Follow-Up of HA

### 3.1. Initial Imaging Assessment

As for preoperative planning, radiographic follow-up includes an anteroposterior pelvic view, ideally standing, and an anteroposterior and profile (urethral, surgical, or Lequesne) hip view, ideally in the supine position, covering the whole material. The Lequesne view offers the possibility to measure cup inclination. Radiographs are performed for the standard follow-up and in case of complications, which may require further imaging techniques. CT is indicated in case of normal or equivocal radiographic findings in case of painful hip, for the assessment of osteolysis when revision is considered, for the evaluation of periarticular masses, fluid collections, and soft-tissue ossifications, and to measure component placement [37]. MRI is mainly used in case of complications, especially for imaging soft tissues around the prosthesis, but its place is not clearly defined yet [37].

### 3.2. Technical Aspects

#### 3.2.1. Computed Tomography

CT acquisition should extend from a few centimeters above the acetabulum to at least 1 cm distal to the tip of the femoral stem (or to the femoral condyles if implant positioning must be checked) with the lowest thickness slice possible and a FOV of approximately 32 cm, concerning the hip. The patient should be placed with a lower limb extended, without pelvic version or rotation, without hip flexion, and foot neutrally rotated [54]. Multiplanar reconstructions must be provided in soft tissue, prosthetic, and bone kernels, with and without metal artifact reduction (MAR) techniques [55]. MAR and native images should both be analyzed as algorithms might change bone and metal appearance (Figure 5). In case of bilateral prostheses, one hip can be elevated during the acquisition to minimize artifacts [54]. Narrower collimation, low pitch (inferior to 1), higher kilovolt peak, and milliampere seconds value improve image quality [37]. At the author’s institution, a single volume is acquired starting 3 cm proximal to the acetabular roof, with the parameters acquisition determined by the patient’s body mass index: tube rotation time 0.75–1 s, 120–135 kVp, 100–450 mAs, slice thickness 0.5 mm, FOV 32 cm, and matrix 512 × 512. In case of infection or pseudo-tumor suspicion, iodinated IV contrast media is used for a second acquisition approximately 2 min after injection, allowing a subtraction reconstruction to assess better contrast uptake (subtraction of the non-contrast-enhanced (CE) images from the CE images). Arterial and or venous-angio-CT (subtracted CT angiography if available) can also be practiced in case of suspicion of vascular pathology [56]. 

#### 3.2.2. Magnetic Resonance Imaging

MRI used to suffer from metal artifacts but has become a part of the routine workup for patients with THA in many institutions. A pelvic acquisition with a large FOV can be acquired with a body coil or a multichannel surface body coil system. Then, a hip acquisition with a small FOV can be obtained with a multi-channel surface coil, a two-part shoulder coil, or a wrapped coil, extending from the anterior to the posterior skin surface, transversally from the pubic symphysis to the skin surface, and craniocaudally from above the acetabulum to the distal end of the prosthesis [57]. To minimize metal artifacts, the amplitude of the gradient must be increased. Therefore, one should prefer a 1.5 T scanner to a 3 T one (as susceptibility artifacts are proportional to the magnetic field strength) and fast spin-echo (FSE) to echo-gradient (EG) sequences, using high receiver bandwidth and thick sections [57]. Adjusting the direction of the frequency-encoding gradient along the axis of the prosthesis can also diminish artifacts [11]. Slice encoding metal artifact correction (SEMAC) (Siemens Healthcare, Erlangen, Germany) and multi-acquisition variable resonance image combination (MAVRIC) (GE Healthcare, Waukesha, WI, USA) have been shown to reduce metal artifacts and improve the depiction of the synovium and bony interfaces with the implant. Notably, isotropic MAVRIC sequences improve the signal-to-noise ratio, conspicuity of lesions, synovium, and periprosthetic bone depiction with less blurring than conventional MAVRIC sequences, at the price of a slightly longer acquisition time [39]. A MAVRIC-SEMAC fusion sequence, MAVRIC SL, can acquire proton density, T1 weighted, and STIR images [58]. To improve image quality, an intermediate echo time should be used to obtain fluid-sensitive images, so are a large matrix in the frequency direction (e.g., 512), a high number of excitations, and an inversion-recovery fat suppression (e.g., STIR) [57]. 

Practically speaking, according to the Radiological Society of North America, intermediate-weighted FSE sequences with a high spatial resolution (periprosthetic bone and soft tissues analysis) and STIR or fat-saturated heavily T2-weighted sequences (fluid and bone marrow edema depiction) should be used [13,57]. T1-weighted and CE sequences are not always recommended. European teams usually perform T1-weighted sequences, and so does the author’s team. The French Society of Musculo-Skeletal Imaging (SIMS) proposes a 3D-T1-weighted MAVRIC sequence (FOV 40 cm, matrix 352 × 352, slice thickness 1.4 mm) or axial and coronal T1-weighted images (MAVRIC or SEMAC), then axial and coronal ± sagittal STIR images (FOV 46 cm, matrix 3384 × 362, slice thickness 3 mm), for a total acquisition time of approximately 19 min [59], without systematic CE sequences, which should be discussed case by case. 

### 3.3. Implants Positioning and Their Implications

The initial placement of prosthetic components must be checked. A good implant position is mandatory from a functional point of view, but mispositioning can also lead to complications, such as dislocation, impingement, and peri-prosthetic fractures. Therefore, radiologists must assess leg length, acetabular inclination and anteversion, the acetabular center of rotation position, femoral offset (FO), femoral neck anteversion (FNA), and femoral stem position (i.e., varus, valgus, or femoral stem position centered, no displacement over time) (Table 1) [14,31,32,33,37,60,61,62].

Leg length discrepancies should not be superior to 0.5–1 cm, as an excessive difference can affect gluteal and iliopsoas muscles [63,64,65]. In this setting, an EOS-imaging should be performed if available [66].

It has recently been proposed (1) to consider the spine and hip relationship (beyond the scope of this review) and (2) to realize a profile radiographic acquisition in the standing and sitting positions on an EOS system, if available, in the preoperative setting in case of lumbar degenerative pathology, and in case of painful or unstable prosthesis before revision [67].

#### 3.3.1. Acetabular Side

The acetabular vertical center of rotation corresponds to the vertical distance between the center of the femoral head and the trans-ischial tuberosity line, and the horizontal center of rotation to the distance between the center of the femoral head and the teardrop shadow (Figure 6). Those values should be similar to the contralateral hip [32], as a lateralized horizontal center might favor dislocation. 

The frontal acetabular inclination is the angle between the lateral edge of the cup and a transischial tuberosity line in the frontal plane and should be of 40 ± 15°, as a lesser angulation would limit hip abduction, whereas a greater one would increase the risk of dislocation [32] and premature wear. The inferomedial border of the acetabular component should be aligned with the bottom of the teardrop on an AP pelvic view (Figure 7). 

Acetabular anteversion is the angle between the acetabular axis and the coronal plane. It is usually measured on a profile radiograph and corresponds to the angle between the edge of the acetabulum and a perpendicular line to the horizontal plane [32]. It also can be measured on a CT-scan, at the level of the center of the femoral head, on an axial oblique plane perpendicular to the pelvic axis (defined by the middle of the superior plate of S1 and the center of the femoral heads), regarding the transischiatic line [68]. Its value should be 5–25°, as lack of anteversion or retroversion can lead to posterior dislocation or iliopsoas impingement and excessive anteversion to anterior dislocation (Figure 8). 

Sagittal acetabulum inclination corresponds to the angle between the edges of the cup and the horizontal axis in the sagittal plane. Its value should be 35–40 ± 10° (Figure 8) and 52 ± 11°, respectively, in the standing and sitting positions. In the sitting position, a low inclination leads to an anterior impingement between the cup and the neck and a compensative augmentation of hip flexion at risk of posterior dislocation, also favored by a femoral retroversion. Inversely, in the standing position, an excessive inclination might lead to a posterior impingement and a risk of anterior dislocation [67]. 

Bendaya investigated anatomical pelvic (beyond the scope of this paper) and implant positioning measurements in patients with good versus poor THA outcomes. The only significantly different parameter was acetabular implant position, essentially with lower frontal inclination in the poor prognosis patients’ group. Femoral implant and pelvic parameters did not significantly differ between the two groups, but the poor patient’s group showed a higher number of parameters deriving from typical values in the standing position only. Those data underscore the need for a global patient analysis better than correcting single parameters to improve planning for both primary and revision surgery [69]. From a functional point of view, a retroversion of the pelvis occurs from standing to sitting position, which also affects the anteversion and inclination of the acetabular implant (6–10° increase for each parameter) [69]. It has been shown that patients who underwent THA without spine disorder had more changes in acetabular implant orientation due to greater adaptability of the spinopelvic junction than patients who had spine fusion, who might be at risk of posterior dislocation because of less femoral head coverage and less acetabular anteversion. Additionally, the more fused levels, the more acetabular anteversion and inclination decrease (about 1° for each fused group for each value) [70]. Therefore, it has been proposed by Lazennec et al. to evaluate lumbosacral junction using standing and sitting EOS imaging or radiographs in patients with spine fusion in THA planning and follow-up [71,72,73], but this attitude is not yet officially recommended.

#### 3.3.2. Femoral Side

The femoral stem position must be analyzed on a pelvic anteroposterior view and a profile hip view. It should be aligned within the femoral shaft with its tip well centered. Varus mispositioning is associated with pejorative outcomes and can lead to lateral pain with a femoral lateral cortical thickening. Valgus positioning can be tolerated [54] (Figure 7). 

The FO should be like the contralateral hip and measure 41–44 mm. This value must be measured on an anteroposterior hip view in neutral rotation, as rotation might induce significant measure variation. It can also be obtained with the EOS system or CT-scan (a method proposed in Figure 9) [54]. A low FO can lead to limping, mobility limitation, and dislocation by gluteal muscles weakness. A loss of FO negatively influences patients’ satisfaction after THA [74]. A too high FO might induce gluteal muscle pain because of exacerbated tension and PE wear. 

FNA should be 10 to 15° to allow a good hip flexion and can be measured on EOS imaging or CT-scan [75] (by applying the method proposed in part 1 of this paper). An excessive FNA increases the risk of anterior dislocation and ischio-femoral impingement [76]. An insufficient one, or a retroversion, increases the risk of posterior dislocation. It has recently been shown that surgeons should be cautious with the expectation of achieving the femoral stem version of an uncemented prosthesis from the preoperative 3D-CT planning, as the preoperatively measured and planned stem orientation was never achieved in Belzunce’s study (discrepancy of −1.4  ±  8.2 degrees with a 95% confidence interval of (−16.9, 13.8)), also with a 2 mm larger FO than the planned one. The latter, however, remained in a normal range [10].

The prosthetic femoral head should be centered within the acetabular cup or slightly inferiorly located as the PE liner has a thicker superior rim. In case of PE liner, wear of 0.1 mm/year is considered normal [54]. Therefore, if it is located upwards, it indicates PE wear and should raise suspicion for granulomatous osseous resorption (Figure 10). 

Resurfacing arthroplasty responds to specific issues. Even though the acetabular component position does not differ from other types of prosthesis (lateral inclination of 30–50° and anteversion of 5–25°), the femoral component must be placed in a relative valgus position of 5–10° to avoid notching the neck (in case of excessive valgus) and to adequately cover the femoral neck [32] (Figure 1).

Components should not displace over time, and components, screws, or cement should not become fractured. Ensuring the absence of material displacement requires a careful comparison of all available radiographs, including the early post-operative ones. One should know that some types of the prosthetic stem can subside from 1–2 mm, especially superolaterally, but no longer than two years or greater than 1 cm [77].

### 3.4. Normal Imaging Findings

#### 3.4.1. Radiolucent Zones (Radiographs, Tomosynthesis and CT)

With radiographs or CT, the presence of periprosthetic radiolucent zones (RLZ) might be a normal finding if it meets strict criteria. Pathologic or regular, they should be described according to the classification of De Lee and Charnley for the acetabular component and with the Gruen zones for the femoral component (Figure 11). Concerning resurfacing arthroplasty, three zones are described around the peg [78] (Figure 11). RLZ around the metaphyseal stem of a resurfacing prosthesis are often asymptomatic, and a neck narrowing can be found without clinical significance [79].

A thin linear RLZ in femoral zone 1 is a frequent finding at the component–cement interface; it results from a lack of contact between these structures during surgery and is thought to be expected if stable over time [32]. An irregular interface between cement and cancellous bone, especially in the greater trochanteric region, and a thin radiolucent line outlined by a sclerotic line parallel to the stem along with the bone–cement interface or around the surface of a cementless component (corresponding to a fibrous membrane) are also considered normal, if non-evolutive [32]. Thin RLZ inferior to or equal to 2 mm are considered normal, but one should know that RLZ superior to 2 mm might be acceptable if stable over time, and RLZ inferior to 2 mm might indicate loosening if they appear during follow-up [37]. Air bubbles into the cement are acceptable and should not raise suspicion of infection if isolated [37].

#### 3.4.2. Adaptative Changes (Radiographs, Tomosynthesis and CT)

Mechanical loading of the hip is assumed by the femoral component in case of HA and transferred distally to the host bone. This results in proximal femoral bone demineralization. Adaptative atrophy also occurs with cementless components in the superomedial acetabulum and in the proximal medial femur within the first two post-operative years and must remain stable. For the same reason, cortical thickening and periosteal reaction at the distal point of the stem reflect successful fixation and appear to be homogeneous and circumstantial in typical cases (Figure 12). Below the tip of the stem, a bone pedestal can occur as a sclerotic transverse line in zone 4, bridging the medullary canal, and can be associated with loosening if it appears or disappears throughout follow-up. Such bone density changes, called “stress shielding”, are secondary to a non-optimal osteointegration and might be asymptomatic or painful (Figure 13). Therefore, follow-up radiographs are mandatory to ensure they do not move into an authentic loosening. In case of uncemented implants, spot welds refer to bone formation from the endosteal surface reaching the prosthesis and indicate stability (Figure 12) [37].

#### 3.4.3. MRI

MRI interpretation is challenging because of an overlap between asymptomatic patients’ imaging findings and clinically relevant ones. The standard post-operative appearance depends not only on the type of implant but also on the post-operative delay. Synovitis, extensive edema or fluid in the implant area, and tracking along soft tissue planes may be encountered in the immediate post-operative time. Over months, soft tissue anomalies tend to regress but may persist along with the surgical incision site or transform into seromas without signs of infection [13]. Signal perturbation related to metal susceptibility often occurs at the superior aspect of the acetabular component and at the femoral stem [13].

The intact periprosthetic cortex and periosteum appear hypointense on STIR and intermediate-weighted fast SE sequences [57]. Complete osseous integration corresponds to direct contact between a sharply demarcated implant or cement and the surrounding bone without separation [57]. A fibrous membrane at the prosthetic interface can be present and manifest as a hyperintense layer of 1–2 mm thickness, suggesting closer surveillance as its effect on implant fixation is poorly known, to make sure it does not move to loosening (Figure 14) [57]. The normal pseudo-capsule should be thin, of low signal intensity, and closely applied to the neck of the implant. Still, a small amount of post-operative fluid without synovitis is standard [57]. In the author’s experience, the pseudo-capsule is not always seen because of artifacts and its interpretation should remain cautious.

Germann et al. investigated normal MRI findings after uncemented THA for two years and found that:Bone marrow edema was frequent all over the femoral stem and in the central acetabular zone at 3 and 6 months after surgery, decreased during follow-up, and sometimes persisted in Gruen zones 1 and 7 overtime, but often in only one area;Inferomedial edema in the acetabulum was infrequent and should raise suspicion for pathology;Periprosthetic bone resorption was frequent during the second post-operative year in Gruen zones 1 and 8 but never thicker than 2 mm;Periosteal edema was shared on the femoral side with a decrease over time, rarely present at two years, and only in non-adjacent Gruen zones, without acetabular side attempt;In the first six months, soft-tissue edema was a constant feature in the surgical access route but never occurred in the second year;Joint effusion was decreasing over time but could be present in the lateral aspect of the joint capsule at two years [19].

On the other hand, one should know that bone marrow edema in the proximal aspect of the stem is frequent in asymptomatic and symptomatic patients, that osteolysis is thought to be more frequent in symptomatic patients in Gruen zone 7, and periosteal reaction is more frequent in symptomatic patients in Gruen zones 5 and 6 [16].

## 4. Complications

HA revision mainly concerns instability (dislocation), loosening, and infection. Other complications include periprosthetic fractures, hardware failure, adverse local tissue reactions, component wear-induced synovitis, tendino-muscular pathologies, heterotopic ossification, and neuropathy. 

Complications can be classified as common to all kinds of arthroplasties (loosening, dislocation, peri-prosthetic fractures, psoas impingement, heterotopic ossification, implant failure, neurovascular and muscle pathology), and specific to bearing surfaces (MoP: wear and osteolysis; MoM: metallosis, pseudo-tumor, and trunnionosis; CoC: squeaking and prosthetic fracture; CoP: ceramic fracture and wear). Resurfacing arthroplasty complications are developed in a separate section. An imaging-based algorithm is proposed in Figure 15 [80,81].

### 4.1. Dislocation

#### 4.1.1. Background

Dislocation can be early (within three months), of good prognostic with a low rate of recurrence, or late (above three months), with high recurrence risk.

Early dislocations, diagnosed on radiographs, might be secondary to post-operative gluteal muscle weakness and/or articular capsule laxity, non-compliance with post-operative restrictions, trauma, and implant mispositioning [19]. They are also favored by a posterior or anterior approach (respectively, posterior and anterior dislocation) [33]. In patients with femoral neck fractures, the use of the posterior approach increases the risk of dislocation, while a non-significant risk is present in cases of dual mobility cup implantation via the lateral approach [82]. In case of single-event dislocation, radiographs are sufficient. It may be treated with “hip precautions” (exercises and positions to avoid) if components are well-positioned on radiographs and proper hip biomechanics are restored. If no trauma or abnormal movement is declared prior to dislocation, muscular weakness can be suspected. On the other hand, if two or more episodes happen, a CT-scan should be performed to determine implant positioning. However, surgery will be required in about one-third of the patients, with a remaining risk of 21–30% of dislocation after revision [83]. Surgical procedures commonly used include increasing femoral head size, correction of implant mispositioning, use of a dual mobility implant, constrained liner, and soft tissue repair, with various efficacities and proper complications (for instance, increased PE wear for larger femoral head) [83]. 

Late dislocations are often multifactorial. The main causes are PE wear, loosening of implants, trauma, non-compliance with post-operative restrictions, trochanteric pseudarthrosis, or amyotrophy. After five years post-operative, soft tissue progressive laxity is also incremented [33]. Primary implant mispositioning is not probable if no dislocation occurred in the post-operative period, but it can be secondary to loosening, which makes implants move into a wrong position, prone to dislocation. The first episode can also be treated with “hip precautions” if trauma or abnormal movements are found in the medical history prior to dislocation and if radiographs are normal, as muscular weakness can also be suspected. In case of recurrent dislocation, even if muscular weakness is possible, a CT-scan is necessary to search for radiographic occult loosening, to evaluate implant positioning and to plan an eventual revision procedure. MRI can also be prescribed to evaluate gluteal muscle and tendons. 

Concerning dual mobility cup, dislocation can be of two types: a loss of contact between the PE insert covering the head and the acetabular piece (classical) or an intra-prosthetic PE dislocation [84], defined as a loss of contact between the head and the PE. The latter can occur during the reduction of a classical dislocation in the early setting and lately secondary to wear [85] (Figure 16).

Dislocation is thought to be related to impingement between bone, implants, or soft tissues, as a dynamic process driven by multiple factors, including hip offset, implant design, component position, and bony geometry [83]. Therefore, evaluating all those factors is mandatory to treat this condition properly. 

Posterior dislocations might be secondary to posterior capsule dehiscence, short external rotator dysfunction, and anterior dislocation to excessive acetabular cup anteversion and anterior capsule lesion [57].

#### 4.1.2. Imaging

Therefore, radiographs and/or CT-scan can be realized to assess implant position, especially looking for an excessive acetabular frontal inclination and an inadequate anteversion, an incorrect FO and/or FNA, or a lower limb length discrepancy. CT also allows for studying gluteal muscle trophicity. MRI is better for studying periprosthetic soft tissues, especially the posterior joint capsule and the short external rotator muscles. An intact posterior capsule is in contact with the greater trochanter but is rarely clearly seen in the author’s experience. A fluid gap indicates failed repair of the posterior capsule and/or muscles, and scar tissue can be seen with high or intermediate signal intensity. Muscle atrophy and fatty infiltration might indicate a nonfunctioning tendon unit [57]. The anterior joint capsule can undergo plastic deformation due to instability and appear thickened, hyperintense, and scarred. CT and MRI may also indicate additional findings after dislocation as component fractures, displacement of the acetabular liner, and persistent dislocation [57]. In the author’s opinion, radiographs and CT are sufficient, and MRI should be realized only in selected cases.

#### 4.1.3. Imaging Perspectives

As dislocation is mostly posterior and occurs during rising from a chair or bending over, frontal and lateral radiographs (EOS imaging or standard) from the spine to the ankles might be performed to assess pelvic alignment and implant position in different functional situations [73,83]. Additionally, Sutphen et al. have proposed an algorithm for recurrent posterior dislocation treatment based on 3D models from CT-scan allowing to simulate of hip kinematics during dislocation (i.e., flexion, adduction, internal rotation) using dynamic modeling software. This procedure allowed to identify bone-on-bone or material–bone impingement in half of the patients, and allowed to properly plan reorientation of the acetabular component, revision of the femoral head to increase hip offset, reorientation of the femoral stem, increasing femoral neck length, and removing impinging bone (i.e., anterior inferior iliac spine and anterior aspect of the proximal femur) to increase hip range of motion. In the other half of their patients, no impingement or limited range of motion were found, suggesting a soft tissue cause [83]. Those data seem to encourage the realization of EOS images or radiographs in different positions and 3D CT-based templating but additional studies are required before recommending their systematic use, in our opinion.

### 4.2. Ergonomics

#### 4.2.1. Background

Prosthetic mispositioning and spinopelvic mobility troubles might lead to femoro-acetabular impingement, pain, wear, and instability. In the majority of revision procedures for unstable prosthesis, Marchetti et al. found macroscopic signs of impingement [86]. Additionally, spinopelvic unbalance can destabilize a hip prosthesis and vice versa, causing a “spine–hip syndrome” or a “hip-spine syndrome” [87]. In the sitting position, an insufficient cup anteversion can lead to an anterior impingement between the cup and the femoral neck with a compensatory hyperflexion of the hip, at risk of posterior dislocation. Contrarily, in the standing position, an excessive anteversion of the cup can lead to a posterior impingement, at risk of anterior dislocation. In case of rigid spinopelvic junction, the cup might be positioned with more inclination and anteversion, but in case of spinopelvic hypermobility, it should be placed with less inclination and anteversion [67]. Some authors recommend anteversion and inclination values in function of the sagittal spinal deformity [88]. 

Even if THA implantation does not narrow the ischio-femoral space, ischio-femoral impingement can occur. It has been recently shown to be associated with a high femoral ante-torsion (approximately of 20°, compared to 15° in a control group). Of note, patients often present non-specific hip pain, so that imaging signs should help radiologists to raise suspicion of ischio-femoral impingement rather than affirm its diagnosis on imaging alone [89].

#### 4.2.2. Imaging

Some mispositionings are obvious on standard radiographs or CT-scans, such as a frankly vertical or horizontal cup, a medialized femoral pivot, or a limb length discrepancy. In other cases, anterior and lateral pelvic and hip radiographs should be realized in the standing and sitting positions (if available with an EOS system imaging), taking into account all the pelvic parameters, prosthesis positioning, and limb length. A CT-scan can also be performed, as it allows to depict an anterior or posterior offset of the cup and to rule out another complication [67].

In the case of ischio-femoral impingement syndrome, CT can be realized to measure femoral ante-torsion, and ischio-femoral and quadratus femoris space. MRI is the best imaging modality, as it can achieve the above-mentioned measurements (respectively, positive if inferior to 15 and 10 mm) and identify edema and fatty infiltration of the quadratus femoris muscle. Abductors’ tears might also be ruled out as they could play a role in this setting [89].

### 4.3. Osteolysis and Loosening

#### 4.3.1. Background

Even though complete osseous integration leads to the highest probability of implant fixation, limited osseous integration might be sufficient to achieve solid fixation without an amount clearly defined [57].

Most of the failures occurring at five years or later result from osteolysis, leading to aseptic loosening and peri-prosthetic fractures. Loosening corresponds to the loss of fixation of a cemented prosthesis or the absence or loss of osteointegration of an uncemented one. In imaging studies, loosening, therefore, corresponds to implant mobilization over time, raising the need to visualize previous radiographs, even the post-operative in the best-case scenario. Infection must be ruled out by punction in case of doubt. Loosening can be “mechanical” (implant mispositioning or miss-dimensioning, poor primary fixation) or “biological”, secondary to almost all kinds of particles released (bone, cement, PE, metal, ceramic) by mechanical wear leading to granulation tissue formation in the osteolytic zones (Figure 17). 

Those osteolytic zones have the appearance of RLZ around the prosthesis at the bone–cement or bone–prosthesis interface, may appear or increase during follow-up, and should be described as mentioned above. In case of cemented prosthesis, the Barrack classification appreciates the quality of the femoral stem cementation [90] (Table 2). 

#### 4.3.2. Imaging

Even though radiographs have been described sufficient for loosening diagnosis, tomosynthesis might increase its diagnostic value and is superior to CT without MAR [55,91]. CT-scan with state-of-the-art MAR better depicts peri-prosthetic osteolysis, especially on the acetabular side [55], and implants complications. CT-angiography of the pelvis should be performed before revision surgery in acetabular piece pelvic protrusion to study the iliofemoral vascular pedicle [54].

#### 4.3.3. Radiographs, Tomosynthesis, and CT

On the acetabular side, the loosening risk becomes higher when the number of RLZ increases (71% in case of 2 zones and 94% in case of 3 zones) [92], and the presence of an RLZ in zone 1 should raise suspicion for loosening and call for close surveillance [80]. In the case of uncemented prosthesis on the femoral side, the apparition of an RLZ of more than 2 mm is always pathologic, except in zone 1 and 7 [80].

Component displacement (i.e., cranial for the acetabular side, distal and varus inclination for the femoral side), pedestal regression, and cement or component fractures are also indicators of loosening, but the only finding indicating definite loosening is excessive component movement, the others being associated with loosening but remaining nonspecific [37]. For those reasons, composite scores have been developed to assess prosthesis stability and fixation [93,94,95] but are beyond the scope of this paper.

Concerning treatment, in cases of acetabular bone defects, the Paprosky classification should be used (Table 3), as it is composed of imaging and surgical findings, and includes treatment recommendations [40]. Before revision surgery, the radiologist must check for acetabular loosening and the amount and type of bone stock loss, the amount and direction of component migration, and liner wear [40]. A 3D-CT reformats whit removal of the femoral head can improve acetabular visualization, in the author’s opinion. Cavitary defects at the acetabular roof and walls, segmental defects of the acetabular rim, lysis of the medial or posterior ischial wall (with or without pelvic discontinuity), and an estimated percentage of host bone in contact with the acetabular cup might be described, as both hip center of rotation and bone stock must be restored. Of note, bone stock loss areas are usually located away from the acetabular rim and are large. In contrast, pre-existing degenerative cysts are smaller and often found at the acetabular roof [40]. Bone grafts or additional acetabular hardware (acetabular cups, rings and cages, metallic acetabular augments) can be used. Revision surgery complications include neuro-vascular damages, infection, fractures, dislocation, bone graft failure, and dislocation of the prosthetic liner [40].

#### 4.3.4. MRI

MRI has gained more and more attention with the amelioration of the sequences previously mentioned. The axial plane would be the most useful for evaluating the bone-implant interface [57]. A hyperintense layer between the host bone and the implant or cement indicates fibrous membrane formation when inferior to 2 mm thickness (Figure 14) or periprosthetic bone resorption when superior to 2 mm and irregular [57]. One must keep in mind that the evaluation of acetabular component fixation is complex because of the convex surface of the implant, exaggerating artifacts [57]. For uncemented prosthesis, STIR hyperintense margin at the metal–bone interface (i.e., osteolysis) indicates loosening (>1.5 mm for the acetabular side and >3.5 mm for the femoral side). Additional contrast enhancement at the metal–bone interface and STIR signal hyperintensity are indicators of prosthetic joint infection (PJI). The magnitude of bone resorption is also an essential factor, as circumferential bone resorption may suggest loosening [57]. Mechanical loosening may be inferred if blood tests for infection and imaging findings for wear-induced synovitis are negative and circumferential osseous resorption is present, as wear-induced synovitis rather induces bulky osteolysis, particularly in case of implant displacement [57]. Therefore, serial MR scans may be necessary to prove loosening [57].

Periprosthetic soft tissue edema can be found in aseptic loosening or PJI. Still, soft tissue anomalies (including edema, abscess, and enlarged lymph nodes of more than 17 mm) and the involvement of both sides of the joint are more frequent in infectious cases. Schwaiger et al. proposed a two-stage approach [38]. First, radiologists should evaluate the presence of signal changes at the periosteum: any signal changes on the acetabular side and the diameter of the signal anomalies adjacent to the metal–bone interface using the cut-offs mentioned above are indicators of PJI or aseptic loosening. Then, soft-tissue abnormalities must be evaluated: enlarged regional lymph nodes and/or both joint sides’ affection might be in favor of PJI. The IV contrast media administration is still debated and might not be indispensable if such findings can be seen on non-CE images.

When conventional imaging techniques are nondiagnostic, bone SPECT-CT can be used as a second-line imaging modality, mainly due to its high negative predictive value, allowing the stopping of investigations in cases of regular exams [96].

The occurrence of primary bone or soft-tissue neoplasms at the site of HA is rare. Still, one must keep in mind that soft tissue tumors are more frequent than osseous and that a mass arising from the bone with bony destruction and extension in the soft tissue or a soft-tissue mass invading the bone is more likely to be a tumor than a “simple” osteolysis from loosening [57]. Differential diagnosis might be challenging when the mass is adjacent to the synovium.

### 4.4. Infection 

#### 4.4.1. Background

There are four types of PJI: positive intraoperative cultures, early post-operative infection, late chronic infection, or acute hematogenous infection [89]. No test offers great sensibility or specificity for diagnosing PJI. The diagnosis of PJI is based on a combination of clinical findings, laboratory evaluation of blood and synovial fluid, and intraoperative findings [97]. The gold standard remains articular fluid aspiration and culture. A recent study showed that almost 30% of PJI were culture-negative after intra-operative sampling. In those cases, next-generation sequencing showed infection in 66% of the cases, which was polymicrobial in 91% [97]. Blood tests are often negative and should not be mistaken for aseptic loosening before articular fluid analysis [97]. 

Significantly, imaging findings might range from standard to frank bone destruction, mimicking loosening or particle disease [33]. Several definitions of PJI exist, but imaging techniques are lacking in most scoring systems. For example, the Musculoskeletal Infection Society has established a scoring system, not considering imaging techniques, that requires at least two positive cultures of the same organism (even in case of commensal micro-organism) or sinus tract with evidence of communication to the joint or visualization of the prosthesis to diagnose PJI, or composite scores with pre- and peri-operative items [98]. 

Nuclear medicine techniques (beyond the scope of this paper) and MRI are more specifically concerned with the imaging workup of this condition [36], but in the authors’ experience, a CT-scan is often required in the first place by the referring physician. 

#### 4.4.2. Septic versus Aseptic Loosening Imaging Signs

Nonetheless, septic loosening is faster than aseptic using previous radiographs and can be associated with femoral periosteal reaction and soft tissue collections [33]. In the author’s institution, when a joint-fluid aspiration is required, a CT-arthrogram is realized at the same time to depict better sinus tracts and loosening zones, often opacified by contrast media. 

MRI signs have been described above and mainly concern synovitis (more precisely lamellation of a thickened hyperintense synovium), extracapsular tissue and bone edema and enhancement, extracapsular collections and sinus tracts, osteolysis, and lymphadenopathy [57]. According to Guerini, the association of a triple hypersignal (i.e., intra-osseous at the contact of the prosthesis, cortical, and peri-osseous in the soft tissues) is of great significance for infection [59] (Figure 18). Galley et al. compared patients with PJI and controls who underwent MRI at least six weeks after THA and found that periosteal reaction, capsular edema, and intramuscular edema were more frequent in PJI [18]. In addition to conventional MRI features, Albano et al. stated that lymph nodes indices, especially concerning their number, compared between the affected and the unaffected side, might be biomarkers of THA infection [99].

One should be aware of the aspect of temporary cement spacer, impregnated with antibiotics, temporarily used when revision surgery is performed to allow functional hip movement while locally treating an infection (Figure 19). Dislocation, periprosthetic fractures, and secondary infection can occur [100].

### 4.5. Synovitis

#### 4.5.1. Background

Joint fluid can be seen either on MRI or CT with metal-artifact reduction, but MRI is more useful in detecting, characterizing, and defining synovitis [57]. 

#### 4.5.2. Classification and Contribution of Imaging Methods

**Nonspecific synovitis** appears as a simple joint fluid of uniform fluid signal intensity with a thickened synovial wall lining, but its clinical importance in the absence of symptoms is unknown [57]. In capsular disruption, fluid might extend into the greater trochanteric or iliopsoas bursae.

**PE wear-induced synovitis** is secondary to contact pressure between the PE and the femoral head, leading to its wear and intra-articular particle release [6]. On imaging, it manifests as the expansion of the hip pseudocapsule by thick and particulate-appearing synovitis of low to intermediate signal intensity, often isointense to muscles, that might communicate and extend into other bursae. Extra-articular deposits from capsular decompression might result in pseudo-tumors formation, resembling those associated with adverse local tissue reactions of metal debris. Bulky osteolysis is often present, with particulate debris in periprosthetic trabecular bone [57]. CT allows a good depiction of osteolysis and osseous granuloma formation, but MRI is superior in soft tissue involvement analysis. MRI should also be interesting before revision surgery in case of frank PE wear for “soft tissue mapping” or if a stable or tiny granuloma depicted on other imaging techniques does not seem sufficient to explicate the patient’s symptoms [59].

**When a histologic diagnosis is available, adverse local tissue reaction** (also called pseudo-tumors or aseptic lymphocytic vasculitis-associated lesions) corresponds to arthroplasty-related metal products, including metallosis caused by metal debris, reactive tissue inflammation to metal ions, and corrosion products, or a combination of those. MRI plays a key role in diagnosing this condition, which must be precocious because of the aggressiveness of soft-tissue destruction. Implant wear contributes to metallic products in joint fluid and periprosthetic soft tissues in well and mispositioned implants (abrasion and corrosion, edge loading secondary to mispositioning, neck-on-cup impingement). Although metal ion levels alone should not be relied on as the sole parameter to determine revision surgery, serum cobalt level of >1 ng/mL and a Cobalt/Chromium ratio > 2 thresholds are thought to be associated with adverse local tissue reaction in MoP THA [101]. Those values were, respectively, 2.8 and 3.8 for dual modular taper THA in Kwon’s study [102].

*Metallosis* results from the shedding of metallic debris (secondary to a MoM prosthesis with corrosion, a conflict between a metallic acetabular cup and the prosthetic neck, or a contact between a metallic head and an acetabular metal back in case of PE wear or dislocation) that induces synovitis and an indolent pattern of osteolysis, potentially leading to loosening (i.e., potentially looking similar to osteolysis and PE wear). Synovitis may contain low-signal intensity or metallic density debris, causing MRI artifacts and bone erosion, best depicted on CT-MAR (Figure 20). Such debris might also be located in periprosthetic soft tissue and lymph nodes [54,57]. Metallic debris presence might also accentuate PE wear (i.e., third fragment wear) [54]. Of note, high serum metal-ion levels can be found in symptomatic and asymptomatic patients and would be associated with pseudo-tumors, so that such a biological finding should lead to the prescription of an MRI to rule out a pseudo-tumor even in asymptomatic patients [6,103].Additionally, referred to as *trunnionosis*, trunnion corrosion corresponds to a soft-tissue reaction to metal debris released from micromotion and mechanical wear at the head–neck or neck–stem junction of modular MoP HA. On MRI, it manifests as an adverse local reaction associated with medial calcar resorption [11,104].

#### 4.5.3. MRI Focus

MRI is the best imaging technique for monitoring adverse reactions to metal debris [12,105]. Its findings range from expansion of the pseudo-capsule with homogenous joint fluid to amounts of synovial proliferation, potentially extended into bursae, and debris causing pseudo-tumors. Peri-synovial soft tissue edema and nodal pathology can also be present and call to rule out infection [57]. A synovial thickness superior to 7 mm and a mixed solid–cystic synovial pattern is thought to be the best predictors of moderate or severe adverse local tissue reaction [106]. MRI findings of metallosis and local adverse tissue reaction may coexist [57]. Three pseudo-tumor grading systems have been described by Anderson [107], Matthies [108], and Haupftfleisch [109], but using MRI with MAR techniques, all showed limited interobserver reliability [110]. In the author’s country, the Hauptfleisch classification is the most used: type I (cystic lesion with a thin wall), type II (cystic lesion with a wall thicker than 3 mm), and type III (predominantly solid lesion containing necrosis and metallic debris) [109], the severity of the symptoms and the revision rate becoming higher from type I to III. Cystic lesions might be challenging to differentiate from infectious lesions, especially in the Anderson classification [85,107].

### 4.6. Psoas Impingement

#### 4.6.1. Background

In the early post-operative months, during the physical activity recovery, pain secondary to impingement between psoas muscle-tendon unit and the uncovered acetabular can occur. This impingement leads to tendinous lesions and can be accompanied by local bursitis, inconstant and nonspecific. Blood tests are negative, and no infectious symptoms are present. As the psoas bursae and the joint capsule might communicate [111,112], care must be taken not to overlook an articular pathology, especially in cases of PJI. 

#### 4.6.2. Imaging

Imaging purposes are to rule out loosening in the first place, then to confirm the diagnosis, and to look for anatomic predisposition. 

Radiographs can show an oversized acetabular or femoral head component, a lack of acetabular anteversion, a screw in the projection of the iliopsoas tendon, an anterior acetabular offset, or an insufficient acetabular covering on a profile view.

A CT-scan is considered the reference exam. It can easily depict anterior acetabular offset, considered pathologic when superior to 12 mm in the axial plane and usually well seen in the sagittal plane [54]. This offset can be due to a lack of acetabular anteversion or retroversion. Impingement might also result from a screw or a cement leak. One should know that the extension of screws beyond cortical margins can be asymptomatic and, therefore, be cautious when mentioning this finding in the imaging report [57].

Ultrasounds can show the anterior offset and the musculotendinous abnormalities, reproduce the pain, and especially guide a local test injection.

Concerning MRI, the iliopsoas muscle and tendon are best depicted in the axial plane, but the tendon can be obscured by metallic artifacts and, therefore, better seen in the coronal and sagittal planes. In tendinosis, a partial or full-thickness tendon tear can be seen, and atrophy and fatty infiltration can be indirect signs of tendon dysfunction or prior release [57].

### 4.7. Squeaking 

Although its origin is still debated, this phenomenon, occurring in CoC (in 1–21% of the cases) or MoM THA, might be favored by implant mispositioning (i.e., a too verticalized acetabular component, excessive or insufficient acetabular cup anteversion) [113]. A component fracture must be ruled out.

### 4.8. Muscle Pathology

#### 4.8.1. Background

Concerning most of the gluteus medius and minimus, tendinopathy can be favored by a trans-gluteal approach, a too early reeducation, a lower limb length discrepancy, or an excessive FO. Those tendons’ pathologies can cause lateral hip pain, abductor insufficiency, limpness, and anterior dislocation [57]. In the preoperative setting, asymptomatic gluteus medius and minimus pathology diagnosed on MRI has been shown to correlate with inferior 2-year post-operative outcomes [114], so they should be described even in the post-operative setting in the author’s opinion. Additionally, isolated open repair or THA and concomitant repair of gluteal tendon tear have been shown to be safe procedures with high levels of satisfaction at short- to mid-term follow-up (visual analog scale of pain and satisfaction), even though the presence of a full thickness tear was a predictor of worst outcomes in Requicha’s study [115].

The rectus femoris tendon analysis must be careful, especially in the case of the anterior surgical approach.

#### 4.8.2. Imaging

MRI and US are usually performed, can confirm the diagnosis, study bursae, and quantify amyotrophy or fatty degeneration. Imaging findings must be strictly correlated to symptoms, typically occurring in hip abduction with limping, as an abnormal aspect of those tendons is frequent with age. Gluteus medius tendon tears are the more clinically relevant, and the gluteus minimus tendon may be released at the time of surgery and/or denervated, diminishing its attempt clinical relevance [57]. On MRI, in the acute setting, peritendinous edema can be found [57] and help to promote tendon pathology to explain patients’ symptoms, as partial tears are frequently encountered in clinical practice in symptomatic and asymptomatic patients. 

MRI can depict a surgical approach, and Agten et al. showed that anterior and direct lateral approaches resulted in less muscle and tendon damage than the posterior and direct lateral approaches [116]. The anterior approach results in less muscular atrophy than the others [117], even though the posterior approach seems to be the most frequently used [118], and some authors do not find differences in clinical outcome scores at one year between the different methods [117]. Wang et al. showed, using 3D MRI images, that the posterior approach seriously damages external rotator muscle and function and that effective muscle repair is beneficial to the muscular morphological insufficiency, therefore calling for an effective valuable repair of the external rotators to improve early post-operative recovery [118]. Via the posterior approach, the piriformis might lose half of its volume, and the root of the internal obturator be damaged, the latter being involved in pelvic organ support and urinary incontinence [118]. 

### 4.9. Neurovascular Complications

The superior gluteal nerve innervates the gluteus minimus and medius and the tensor fascia lata muscles. Postsurgical damage of this nerve should be suspected in case of atrophy of those abductor’s muscles, especially in limps. In contrast, fatty atrophy of the anterior gluteus minimus fibers with an intact tendon is often found in asymptomatic patients due to selective denervation without clinical relevance. On the other hand, fatty atrophy of the posterior fibers of the gluteus minimus and medius should raise suspicion of a tendon tear [119]. 

In the immediate post-operative period, edema might involve the sciatic nerve and irritate it but typically resolves over time. An unexpected evolution should raise suspicion for an impingement related to hardware malposition or collection. Deep vein thrombosis related to hardware may also occur [13]. 

In the chronic setting, neurovascular complications are often related to synovial expansions, and periprosthetic neuroma may be detected by MRI [13].

### 4.10. Peri-Prosthetic Fractures and Stress Reactions

#### 4.10.1. Background

With an overall incidence of 18%, they occur during component implantation or after surgery and are favored by periprosthetic bone resorption, osteolysis, implant loosening, osteoporosis, femoral stem varus positioning, and trauma [57]. 

Mainly concerning the femoral side, fractures are classified according to the Vancouver classification, based on the location of the fracture, the amount of available proximal bone stock, and the stability of the stem. Type A fractures are peri-trochanteric fractures (subtypes: AL = lesser trochanter and AG = greater trochanter). Type B fractures occur around or just below the tip of the stem (subtypes: B1 = well-fixed stem, B2 = not-well-fixed stem, B3 = poor bone stock in the proximal femur and not-well-fixed stem). Type C fractures occur below the femoral stem. These fractures can be peri-operative during femoral stem placement (especially for uncemented prosthesis), secondary to minor trauma, or varus positioning. They need to be detected as they can require further treatment. 

Acetabular fractures are rare and usually result from trauma or osteolysis. Radiographically occult acetabular fractures can occur during cup fixation, especially at the superolateral wall, but do not require further treatment if secure fixation has been confirmed during surgery [120]. Acetabular, femoral, or pelvic stress fractures are rare, difficult to diagnose on radiographs, and justify a CT and/or MRI pelvic acquisition whenever they are suspected [54,57].

Trochanteric pseudarthrosis can result from revision procedures or technical difficulties, is favored by gluteal muscle tension, and lead to a risk of dislocation. On imaging, they might appear as metallic wires rupture, fracture, and trochanteric ascension and fragmentation [54]. Femoral neck fractures only occur in resurfacing arthroplasty [33].

#### 4.10.2. Imaging

The above-mentioned findings are variably depicted with radiographs but are clearly visible with CT-scan in the author’s experience, which is often required for pre-operative planning.

On MRI, stress reactions are localized signal hyperintensity of the marrow cavity and endosteum, thickening and hyperintensity of the cortex and periosteum without fracture, and adjacent soft-tissue edema (Figure 21). Marrow signal hyperintensity can be a normal finding after several months, secondary to implantation of surgical technique. In this setting, the lack of periosteal reaction and the typical appearance of soft tissues help to make it commonplace [57].

### 4.11. Heterotopic Ossification

In up to half of the patients during post-operative weeks, they correspond to the formation of new lamellar bone within periprosthetic soft tissues. During the osseous maturation phase, local pain and swelling with body temperature may occur and be difficult to differentiate from infection. On MRI, immature bone manifests as heterogeneous processes with mass effect on the surrounding tissue, but CT easily depicts mineralization [57]. On radiographs, the Brooker classification ranges from stage 1 (small ossifications) to stage 4 (bony ankylosis); CT and MRI are helpful for precise anatomical relationships with soft tissues and vascular structures [54].

### 4.12. Implant Failure

The femoral stem can break, its modular component dissociate, and its sintered beads shear off (opaque micro-fragments separated from the porous-coated femoral stem) [33]. Additionally, the acetabular liner can wear, break, or dissociate. 

Prosthetic fractures are rare, secondary to misleading conception or trauma, essentially concern CoC HA (head or acetabular insert), and require careful analysis of the components and of their environment as they can lead to particles reactive synovitis [54].

Still concerning CoC prosthesis, femoral head fracture is secondary to trauma or hyperactivity [85] and leads to functional impairment, whereas acetabular fractures are rare, non-secondary to trauma, and might become symptomatic only in case of displacement [13,113,121]. Acetabular fractures may be favored by excessive anteversion and poor implantation of the insert in the metal back [85]. Ceramic debris might lead to osteolysis and need to be carefully overlooked as it must be resected in case of revision before a new prosthesis becomes implanted (Figure 22).

As acetabular bone loss is a significant challenge in HA revision surgery, all the components used to reconstruct an acetabular socket might be carefully analyzed, as local constraints might favor their breaking and displacement. In the author’s opinion, CT offers the best depiction of implant integrity.

### 4.13. Resurfacing Arthroplasties

Those prostheses, usually composed of cobalt/chrome components, can be considered apart. Complications are rare and mainly related to suboptimal surgical procedures leading to impingement or femoral neck fracture [3]. Uncomplicated components should have an abduction of about 40° and an anteversion of about 20° on the acetabular side, and a neutral or slightly valgus position on the femoral side. No notch should be seen at the neck. Minor RLZ around the femoral stem is normal. Femoral neck thinning adjacent to the femoral prosthesis is frequent, and about 70% of patients show narrowing without clearly defined significance [3]. 

A femoral neck fracture is an early complication. Infection and loosening rates are poorly known but thought to be rare. Dislocation is very rare. Mispositioning can lead to impingement and pain on the acetabular side. An excessive femoral piece valgus position exposes to the risk of notching, and an excessive varus to the risk of fracture (Figure 1). Pseudo-tumors occur in approximately 1% of patients [3].

## 5. Limitations

This paper brings a comprehensive and extensive literature review concerning THA imaging workup, concerning all imaging studies and implants design. However, to date, no consensus exists concerning the optimal imaging diagnostic strategy, and we are not able to bring clear evidence-based recommendations. After radiographs, imaging studies are often prescribed depending on the availability of each imaging technique in each institution and depending on the referring physician’s habits. Concerning asymptomatic patients’ follow-up, it is impossible to state that EOS imaging (if available) could replace radiographs. Nor can we recommend systematical standing and sitting radiographs in case of asymptomatic or symptomatic prosthesis. The use of MRI is growing, especially with MAR sequences, and for soft tissue analysis, but it is still considered a procedure that “may be useful to evaluate hip arthroplasties with suspected soft-tissue or periprosthetic abnormalities” by the American College of Radiology, which does not recommend MRI as a first intention procedure [122]. CT is readily available and has good diagnostic performances. In our experience, it is the best imaging technique to assess mechanical complications when considering implant integrity and osteolysis, but still exposes patients to ionizing radiations and IV contrast media, the latter being less frequent with MRI. However, depending on the local technical conditions, the clinical suspicion, and the patients’ comorbidities, different imaging strategies might still be applied. In this context, one should be aware of each technique’s advantages and drawbacks, to improve patient care.

## 6. Conclusions

HA follow-up requires systematic and standardized radiographic follow-up, even in asymptomatic patients, to assess bone stock and implant positioning. First, the type of HA, its bearing surfaces, and fixation method must be recognized. Then, its position and the bone–prosthesis interface must be compared to previous radiographs if available to detect silent complications in case of systematic follow-up. Normal imaging findings, including radiographs, CT-scans, and MRI, might not be mistaken for complications and should be interpreted with caution and correlated to the clinical context. In cases of complication, a CT-scan used to be the standard reference, but MRI has gained more and more prevalence and technical improvement and might be part of the routine imaging workup in such patients, especially when it comes to soft tissue depiction and infectious loosening. Concerning implant mispositioning, even though CT-scan remains a valuable alternative, standing and sitting position radiographs seem to delineate as a useful tool to assess the patient-specific safe zone and include spine and lower limbs in the referring physician’s reflection when evaluating a painful hip prosthesis, especially in cases of instability. Proper imaging workup and diagnosis relies on acknowledging the type of prosthesis, the post-operative delay, the clinical history, and eventual blood tests. All these elements should be put together, leading to a robust and systematic analysis. Further studies comparing CT and MRI, both with MAR, in case of complications, and radiographs and EOS-imaging for standard follow-up should be realized. Additionally, “CT-like” MRI sequences could be evaluated in HA imaging.

## Figures and Tables

**Figure 1 jcm-11-04416-f001:**
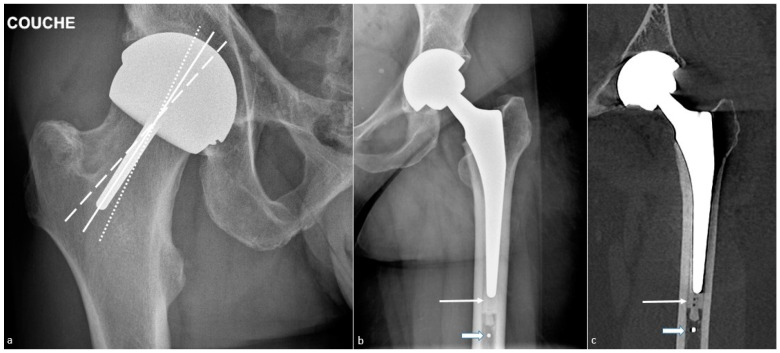
**Total hip arthroplasties.** A resurfacing arthroplasty is shown on (**a**) an antero-posterior hip radiograph and a conventional arthroplasty on (**b**) an antero-posterior hip radiograph and (**c**) a frontal CT-scan reformat. On (**a**), note the neutral position of the stem of the femoral component (white line). One should consider that slight valgus can be tolerated (little-dotted white line) and that varus positioning should not occur (large-dotted white line). On (**b**,**c**), also note the cement (thin arrow) and a metallic marker at the distal part of a cement restrictor (thick arrow).

**Figure 2 jcm-11-04416-f002:**
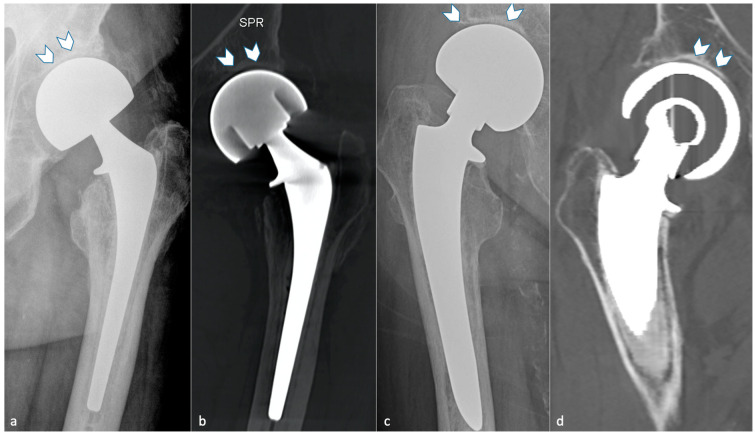
**Hip hemi-arthroplasties.** A unipolar arthroplasty is shown on (**a**) an antero-posterior hip radiograph and (**b**) a coronal slice of CT-MAR. A bipolar arthroplasty is shown on (**c**) an antero-posterior hip radiograph and on (**d**) a coronal slice of CT-MAR. Note that the native acetabulum can be seen on all imaging modalities in both cases (white arrowhead).

**Figure 3 jcm-11-04416-f003:**
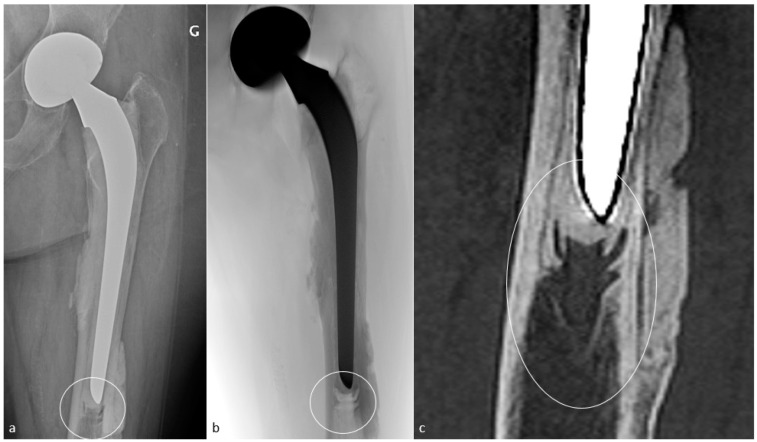
**Distal centralizer in a cemented femoral stem.** A centralizer is shown on an anteroposterior radiograph (**a**), a tomosynthesis (**b**), and a CT-scan reformat (**c**) (white circle). Note its radiolucent and hypodense aspect.

**Figure 4 jcm-11-04416-f004:**
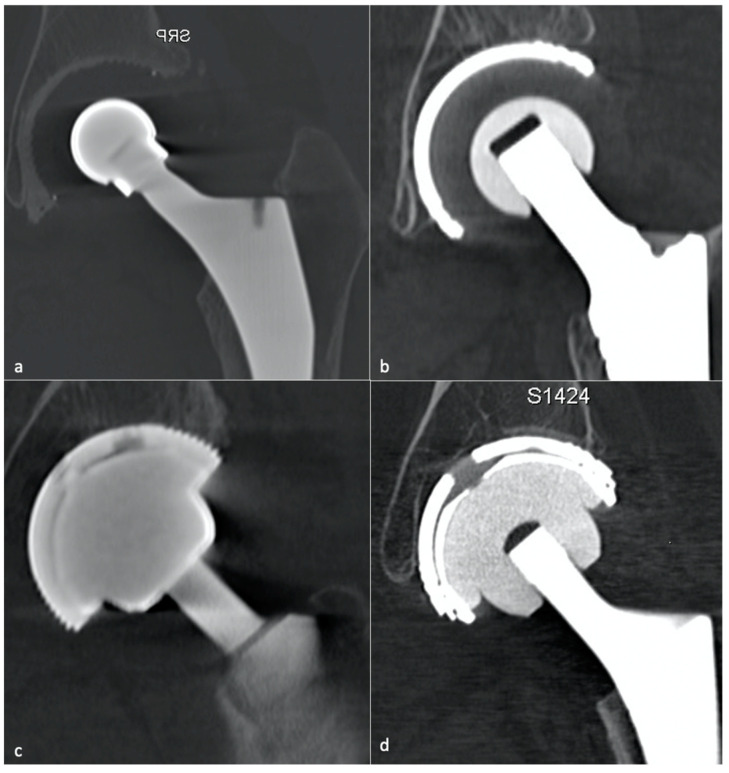
**Bearing surfaces.** Coronal slices of CT-scan showing on (**a**) a metal–polyethylene couple, (**b**) a ceramic–polyethylene couple, on (**c**) a metal–metal couple, and on (**d**) a ceramic–ceramic couple. The material can be recognized by its density with a «visual» scale as polyethylene is less dense than ceramic which is itself less dense than metal.

**Figure 5 jcm-11-04416-f005:**
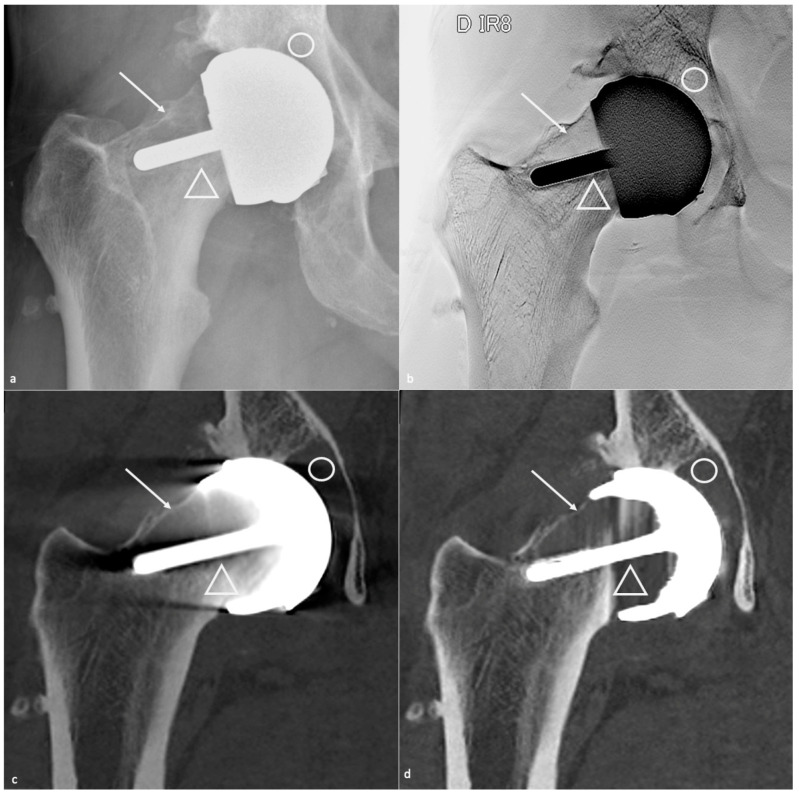
**Peri-prosthetic osteolysis.** Radiolucent zones are shown on (**a**) an anteroposterior hip radiograph, (**b**) an anteroposterior tomosynthesis view, (**c**) a coronal CT image, and (**d**) a coronal metal-artifact reduction CT image. All imaging modalities depict true osteolysis in the superomedial acetabulum (grey circle). Superior femoral neck osteolysis (grey arrow) is doubtful on (**a**,**d**) but is depicted on tomosynthesis (**b**) and CT with metal-artifact reduction (**d**). On the other hand, artefactual osteolysis is seen in (**d**) (grey triangle), underscoring the need for a combined interpretation of all imaging modalities.

**Figure 6 jcm-11-04416-f006:**
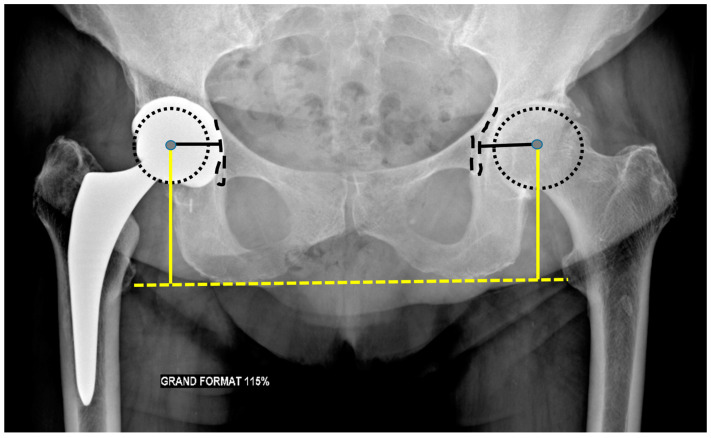
**Acetabular centers of rotation.** Pelvic anteroposterior radiograph. The horizontal center (black line) corresponds to the distance between the femoral head (dotted circle) center (grey point) and the teardrop (large black dotted curve) shadow, the vertical one (yellow line) to the distance between the center of the femoral head and the transischiatic line (dotted yellow line).

**Figure 7 jcm-11-04416-f007:**
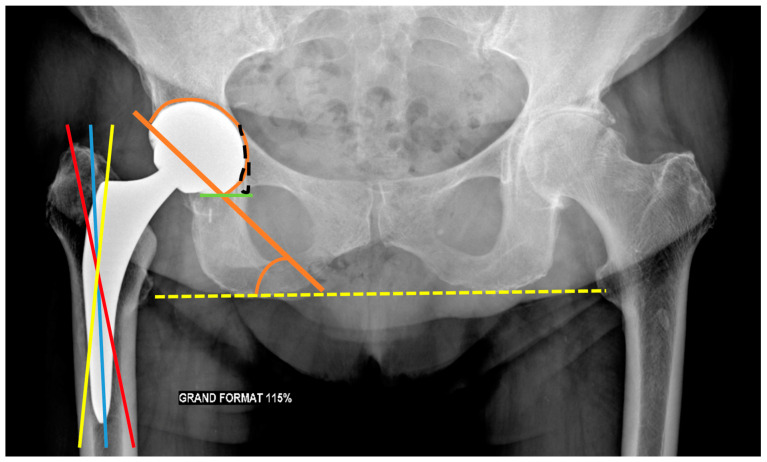
**Acetabular frontal inclination and femoral stem position.** Pelvic anteroposterior radiograph. The acetabular inclination corresponds to the orange angle between the acetabular piece contour (orange circle) and the transichiatic line (dotted yellow line). Additionally, note that the acetabular piece is aligned (green line) with the bottom of teardrop shadow (dotted black curve). The femoral stem should be placed in a neutral position (blue line). A slight valgus (red line) can be tolerated, but varus (yellow line) should not occur.

**Figure 8 jcm-11-04416-f008:**
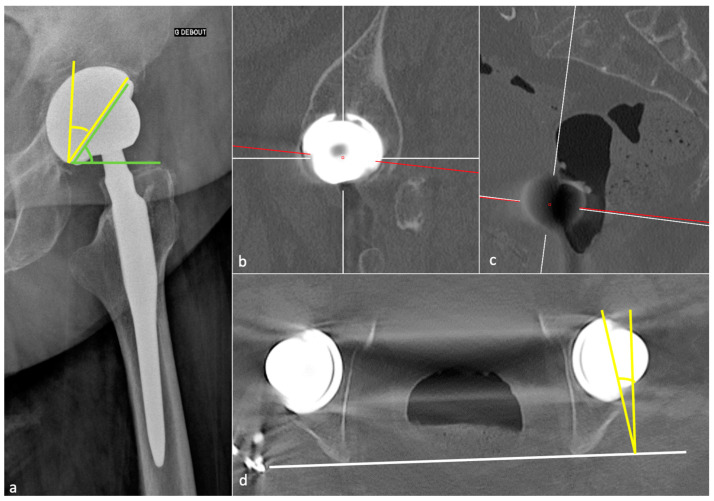
**Acetabular cup anteversion and sagittal inclination.** The radiographic angle of anteversion is shown on (**a**) a hip profile radiograph and corresponds to the angle between the vertical axis and the acetabular piece edges (yellow angle). The sagittal acetabular inclination is shown in (**a**) and corresponds to the angle between the horizontal axis and the acetabular piece edges (green angle). A CT method is then proposed to calculate anteversion: on (**b**) a sagittal CT slice, one should restore the femoral head center, then align it within the middle of S1 superior plate as shown on (**c**) in the sagittal plane. The anteversion can be measured in the so-created plane (red lines) on (**d**), as the yellow angle between the cup edges and the perpendicular line to the transischiatic line (white line).

**Figure 9 jcm-11-04416-f009:**
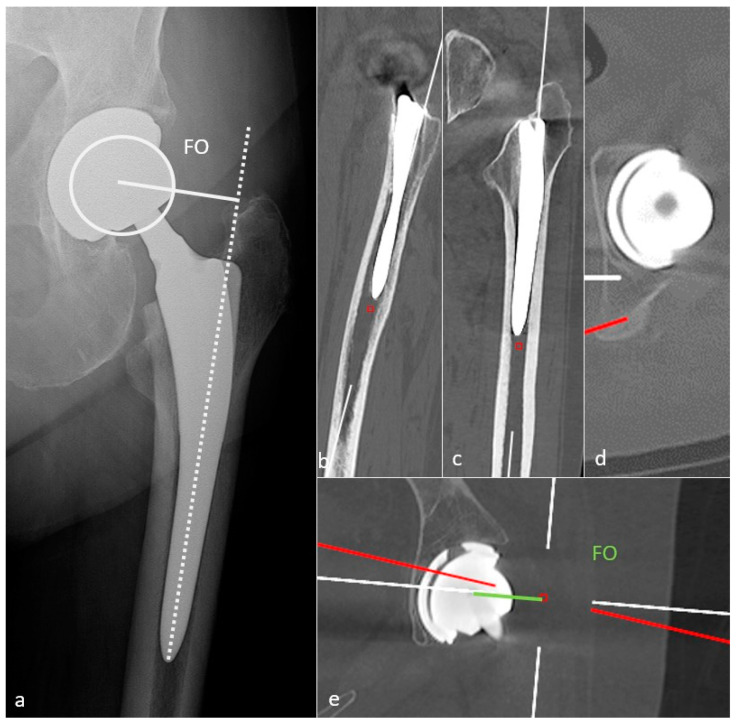
**Femoral offset (FO).** The radiographic method is shown on (**a**) an anteroposterior hip view and corresponds to the perpendicular distance (white line) between the femoral head (white circle) and the femoral axis (dotted white line). A CT method is also proposed. First, on (**b**), in the sagittal plane, the femoral inclination is considered, then on (**c**) in the frontal plane, and the level of the center of the femoral head is repaired in (**d**). Finally, in (**e**), the FO can be measured (green line) as the perpendicular distance between the resulting femoral axis (white vertical line) and the center of the femoral head.

**Figure 10 jcm-11-04416-f010:**
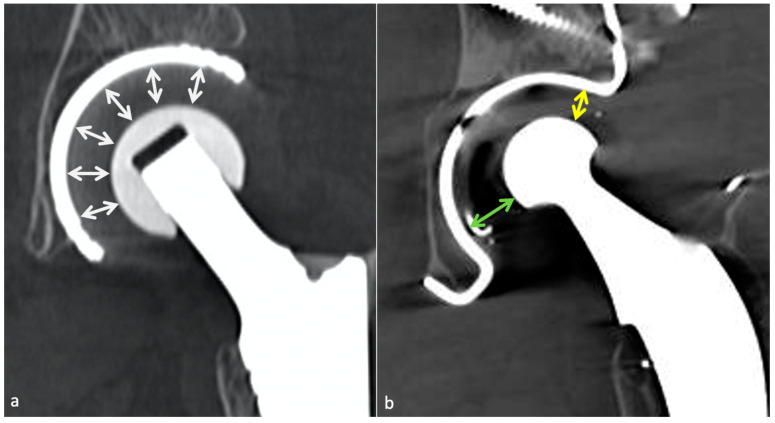
**Polyethylene wear.** Normal polyethylene appearance is shown in (**a**) a frontal CT-scan slice, with a regular thickness as all double arrows are equal, whereas wear is shown in (**b**) a similar frontal CT-scan slice, with asymmetric thickness as the superior yellow double arrow is smaller than the inferior green one.

**Figure 11 jcm-11-04416-f011:**
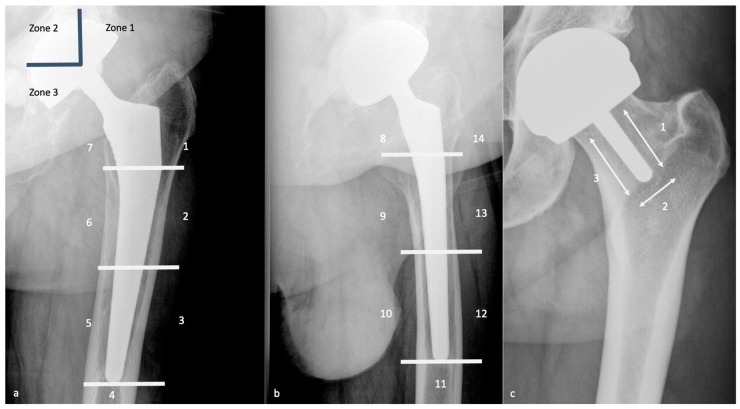
**Acetabular and Femoral Zones to be described for osteolysis.** The De Lee and Charnley acetabular zones and Gruen 1–7 femoral zones are shown on (**a**) an anteroposterior hip view, the Gruen 8–14 femoral zones on (**b**) a profile hip radiograph. The three zones to consider for femoral stem in resurfacing arthroplasties are shown on (**c**) an anteroposterior hip radiograph.

**Figure 12 jcm-11-04416-f012:**
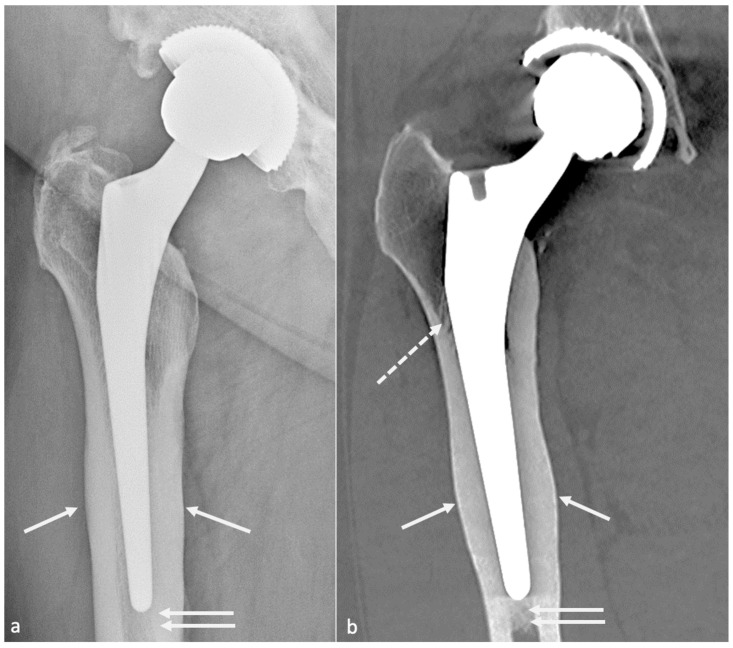
**Successful fixation of an uncemented total hip arthroplasty.** Regular and circumferential cortical thickening (white arrow) and bony pedestal (double white arrow) are seen on (**a**) an anteroposterior hip view and (**b**) a coronal CT-scan slice. Additionally, note spot welds (white dotted arrow) better depicted on CT-scan, and the slight valgus position of the femoral stem, better depicted on the radiograph.

**Figure 13 jcm-11-04416-f013:**
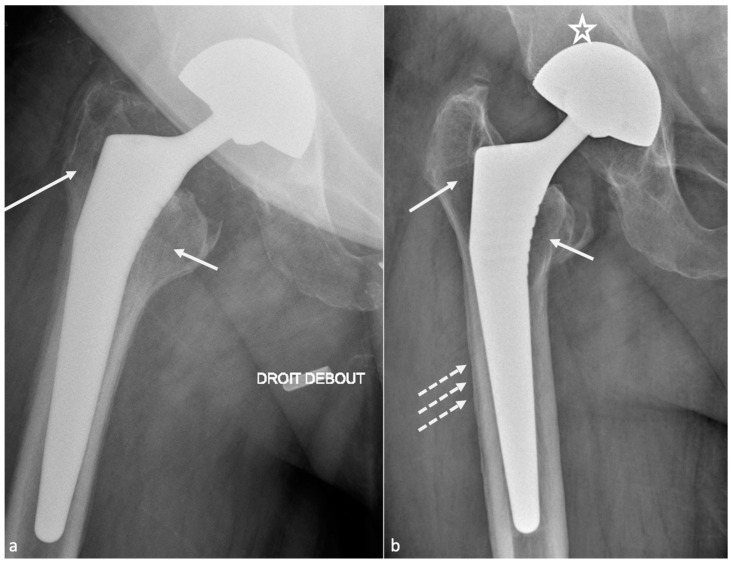
**Stress shielding.** Post-operative (**a**) and one-year follow-up (**b**) radiographs show proximal femoral (white arrows) and superomedial (white star) acetabular demineralization. Additionally, note the subtle lateral unilamellar periosteal reaction (dotted white arrows) on (**b**), which might be favored by the slight varus position of the stem.

**Figure 14 jcm-11-04416-f014:**
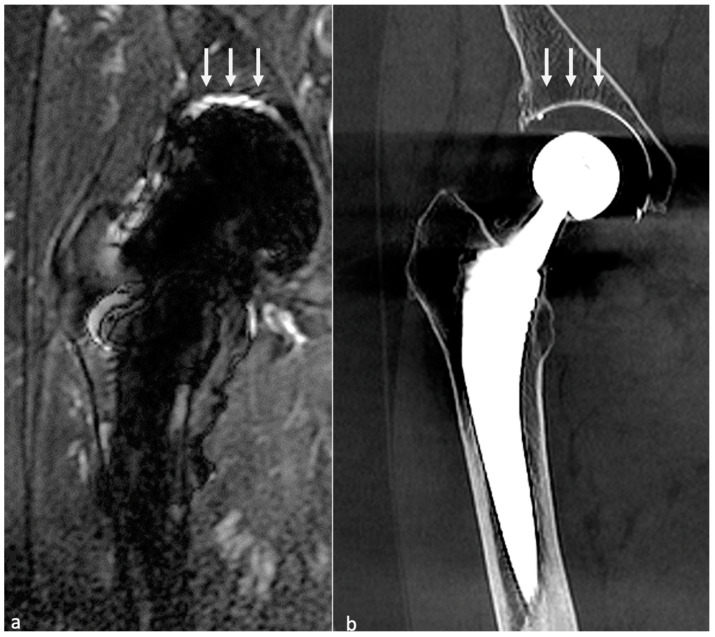
**Acetabular fibrous membrane.** A thin STIR hyperintense layer (white arrows) is shown close to the acetabular piece on (**a**) a frontal STIR image, without osteolysis in the corresponding zone on (**b**) a coronal CT-scan slice.

**Figure 15 jcm-11-04416-f015:**
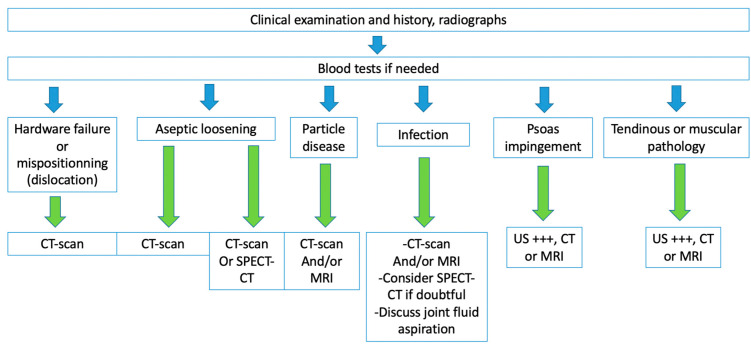
Imaging diagnostic algorithm proposition, adapted from Blum et al. [80,81].

**Figure 16 jcm-11-04416-f016:**
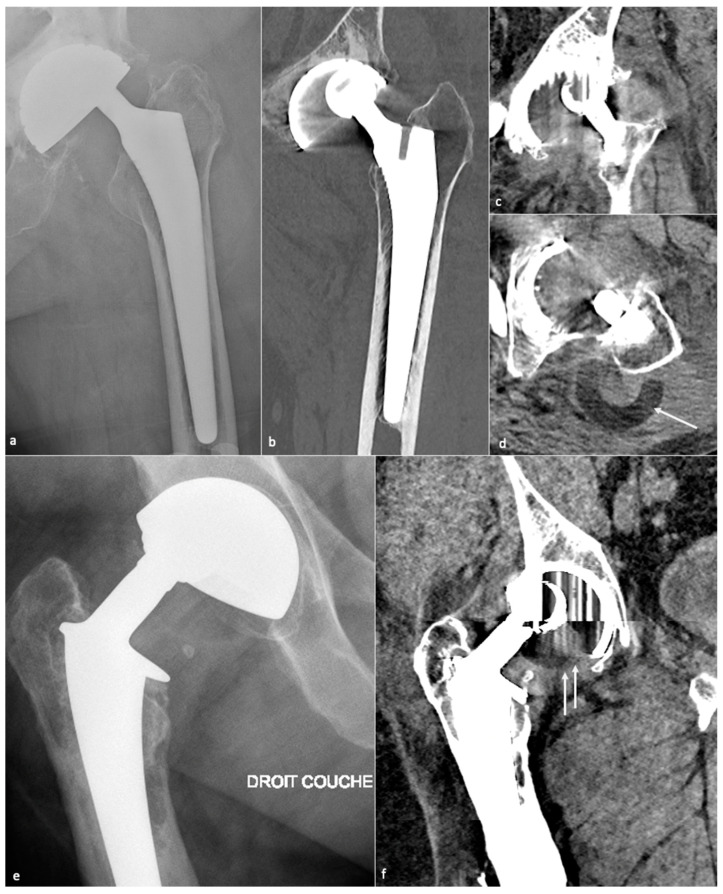
**Polyethylene dislocations.** An intra-prosthetic dislocation is shown on (**a**) an anteroposterior radiograph and (**b**) a coronal CT-scan with bone kernel where out-of-round of the femoral head is seen. Using soft tissue kernel in (**c**) frontal and (**d**) axial CT-scan slices, no polyethylene is seen in its usual position in (**c**) but is depicted behind the great trochanter in (**d**) (white arrow). In another patient, an intra-prosthetic dislocation is shown on (**e**) an anteroposterior hip radiograph where exenteration of the femoral head is seen, and on (**f**) a frontal CT-scan slice where the polyethylene (double white arrow) is located downwards.

**Figure 17 jcm-11-04416-f017:**
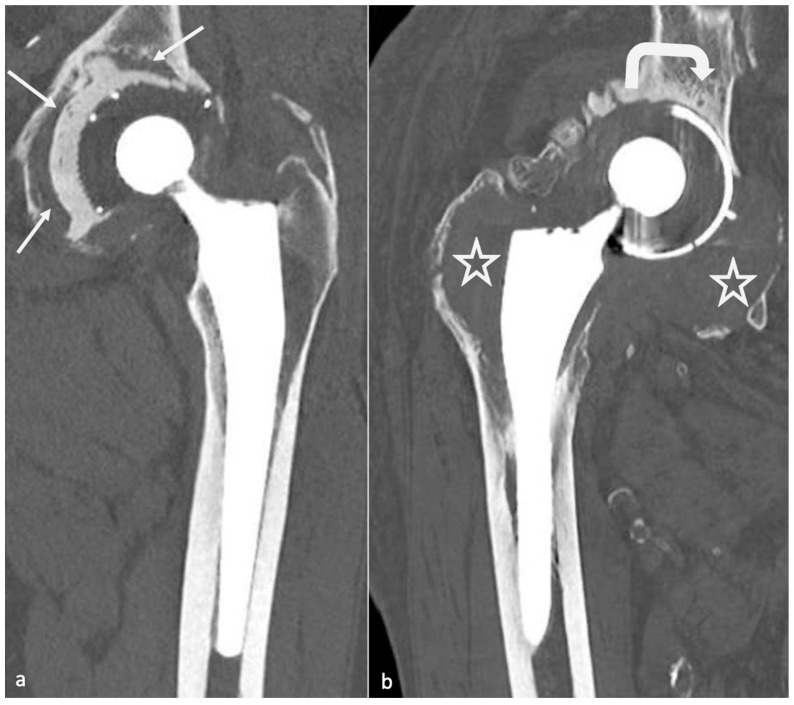
**Mechanical and granulomatous loosening.** An acetabular mechanical loosening is shown on (**a**) a frontal CT-scan slice with regular acetabular osteolysis (white arrows). A granulomatous loosening is shown on (**b**) a frontal CT-scan slice and consists of bulky osteolysis (white stars) both on the femoral and acetabular sides, with a displacement of the acetabular metal-back (white curved arrow).

**Figure 18 jcm-11-04416-f018:**
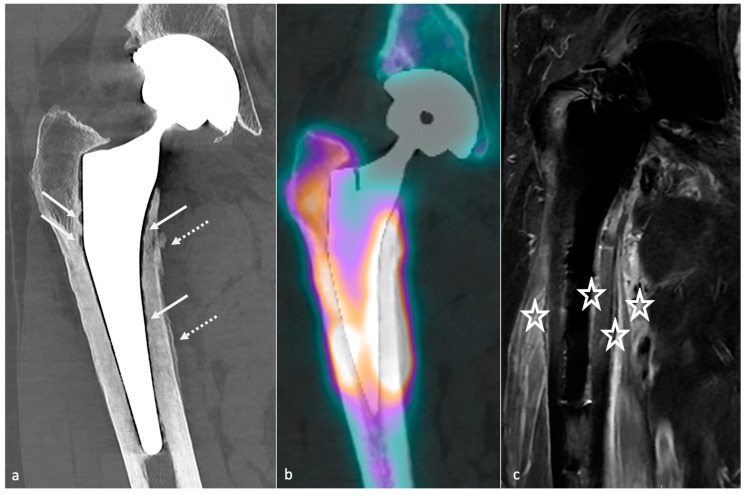
**Infectious loosening (staphylococcus lugdunesis).** An infectious loosening is shown on (**a**) a frontal CT-scan slice with irregular femoral endosteal osteolysis (white arrows) and periosteal reaction (dotted arrows), (**b**) a frontal SPECT-CT slice with diffuse hyperfixation, (**c**) a frontal T2 STIR image with a peri-prosthetic, cortical, and soft tissue hypersignal (white stars).

**Figure 19 jcm-11-04416-f019:**
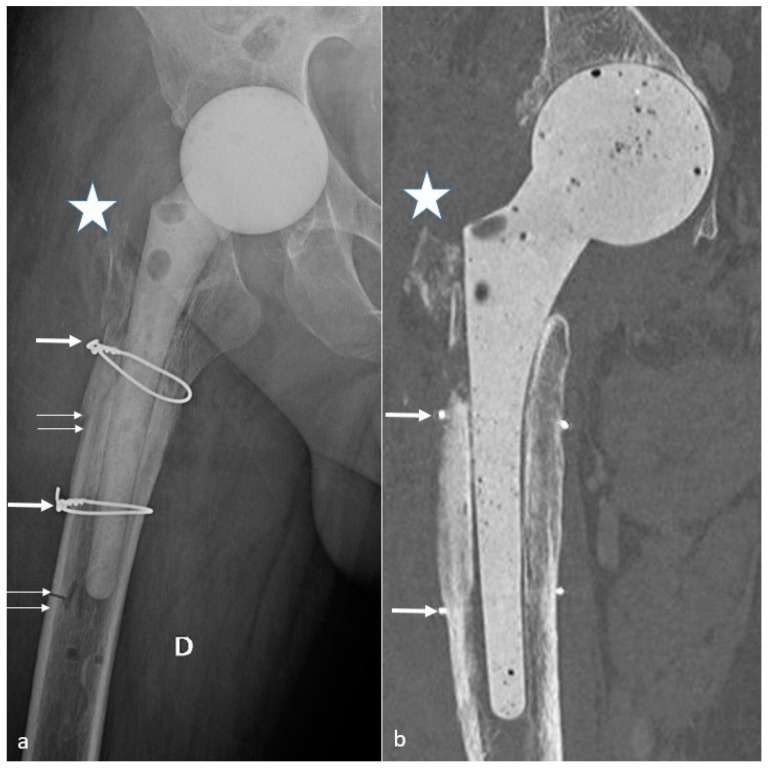
**Spacer appearance.** A spacer is shown on an AP radiograph (**a**) and a coronal CT-scan slice (**b**). Note the cerclage wires (thick arrows), the trochanteric lysis (white star), and the femorotomy sequelae (thin double arrows). NB: “D” in (a) means “droite” and is referred to the right side.

**Figure 20 jcm-11-04416-f020:**
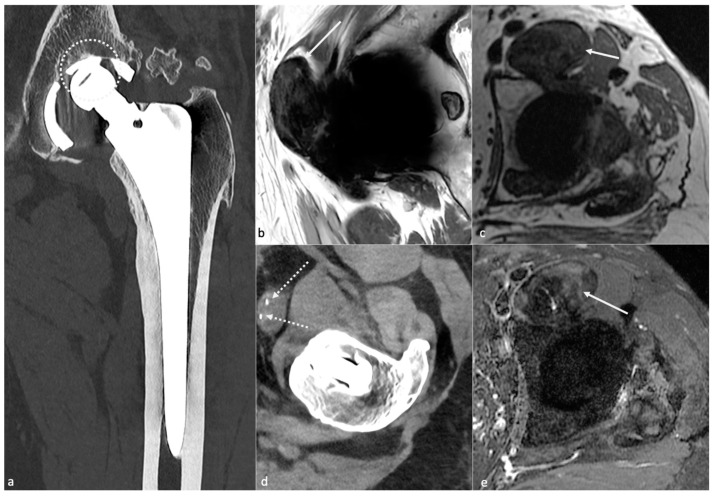
**Metallosis.** Contact between the acetabular metal-back and the femoral head (dotted white circle) is shown on (**a**) a frontal CT-scan slice, secondary to polyethylene wear. A pseudo-tumor extending into the iliopsoas bursae is shown on (**b**) a sagittal T2-weighted image, (**c**) an axial T1 MAVRIC and STIR (**e**) images as hypointense. Additionally, note on (**d**) an axial CT-scan slice the presence of metallic debris (dotted white arrows) into the pseudo-tumor.

**Figure 21 jcm-11-04416-f021:**
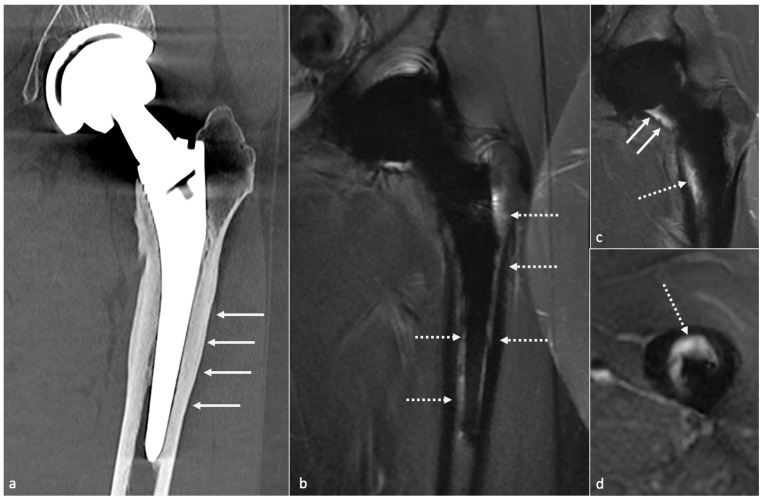
**Stress reaction.** A stress reaction is shown on (**a**) a coronal CT-scan slice with a varus stem position and asymmetric periosteal reaction (arrows). Periprosthetic bone marrow edema (dotted arrows) is doubtful on (**b**) a coronal STIR image but is depicted on another (**c**) coronal and (**d**) axial STIR image. Additionally, note the small amount of joint fluid (double arrow) on (**c**), considered nonspecific.

**Figure 22 jcm-11-04416-f022:**
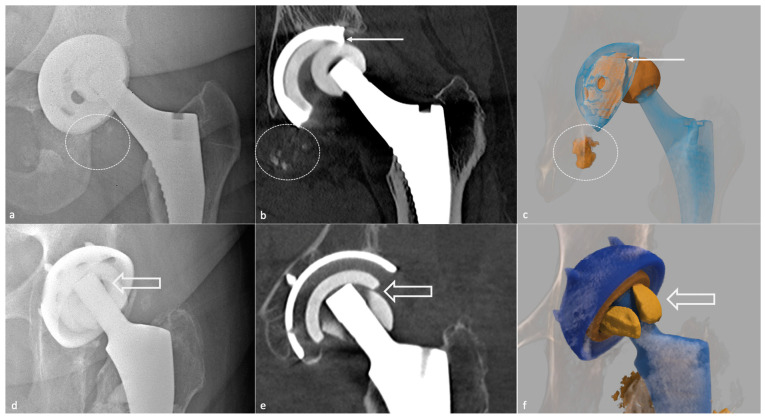
**Ceramic fractures.** An acetabular liner fracture is shown on (**a**) an anteroposterior hip radiograph, (**b**) a coronal CT-scan slice, and (**c**) a global illumination reformat with femoral head exenteration. The fracture is better depicted in (**b**,**c**) (white arrow), and ceramic debris can be seen on (**a**–**c**) (dotted circle). A femoral head fracture in a case of sandwich ceramic liner is shown on (**d**) an anteroposterior hip radiograph, (**e**) a coronal CT-scan slice (thick white arrow), and (**f**) a global illumination reformat. Note the interposition of a radiolucent space between the metal back and the ceramic liner due to the presence of PE, best seen on the CT-scan image.

**Table 1 jcm-11-04416-t001:** Prosthetic implant positioning and their implications.

Measurement	Normal Value	Consequences of Mispositioning
**Leg Length**	< 0.5–1 cm of differences between both sides	Increased discrepancy: gluteal and iliopsoas muscles affection
**Acetabular side**		
**Frontal acetabular inclination**	40 ± 15°	- Decreased: hip abduction limitation
		- Increased: dislocation risk
**Sagittal acetabular inclination**	35–40 ± 10° in standing position	- Increased: posterior impingement, anterior dislocation
	52 ± 11° in sitting position	- Decreased: anterior impingement between the cup and the neck, posterior dislocation
**Acetabular anteversion**	5–25°	- Lack of anteversion or retroversion: posterior dislocation, iliopsoas impingement
		- Excessive anteversion: anterior dislocation
**Acetabular center of rotation position**	Similar to the contralateral hip	Lateralized: dislocation risk
**Femoral Offset**	41–44 mm (or similar to contralateral hip)	- Decreased: limping, mobility limitation, and dislocation by gluteal muscles weakness
		- Increased: gluteal muscles pain and polyethylene wear
**Femoral side**		
**Femoral Stem position**	Neutral or slight valgus	Periprosthetic fracture and stress reaction in case of varus
**Femoral Neck Anteversion**	10–15°	- Increased: anterior dislocation and ischio-femoral impingement
		- Decreased: posterior dislocation
**Femoral Head**	Centered or slightly inferiorly located	- Particle disease if located upwards (wear)

**Table 2 jcm-11-04416-t002:** Barrack classification.

Stage	Radiological Aspect
A	Complete filling of the medullary canal
B	RLZ inferior to 50% of the cement-bone interface
C	RLZ of 50–99% of the cement-bone interface
D	Complete RLZ at the cement-bone interface

RLZ: radiolucent zone.

**Table 3 jcm-11-04416-t003:** Paprosky classification.

Type	Imaging and Operative Findings
**I**	Acetabular rim and columns intact
	Almost complete host bone support of the component
**II**	Superior migration inferior to 3 cm
	Distorted acetabular rim withtout columns attempt
	Host bone support superior to 50%
IIA	Superior and medial cavitation. Intact rim
IIB	Segmental supero-lateral defect (less than 1/3 of circumference)
IIC	Medial wall lysis with acetabular protrusion
**III**	Migration superior to 3 cm
IIIA	Missing bone in the 10 AM-2 PM positions, teardrop lsysis
	Walls compromised
	Columns nonsupportive
	Superior migration
IIIB	Missing bone in the 9 AM-5 PM positions, teardrop lysis
	Walls compromised
	Columns nonsupportive
	Superior or medial migration
**Pelvic Discontinuity**	Fracture line through columns
	Obturator foramen asymetry on AP pelvis radiograph
	Superior and inferior hemipelvis separation

## Data Availability

Not applicable.

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
