# Peer review of "Imaging in Hip Arthroplasty Management Part 2: Postoperative Diagnostic Imaging Strategy"

_jcm, 2022, doi:10.3390/jcm11154416_

Round 1

Reviewer 1 Report

1.      Line 2-3, uppercase and lowercase of the title needs to be revised

2.      In the title of the present manuscript, it has “Part 2”? where is part 1, I have searched for it but have not found

3.      What is the novelty of the present review? Imaging in total hip arthroplasty has been published in several previous works of literature. Based on the reviewer's evaluation the present article does not bring something new and serious scientific contribution which makes it not suitable for the recommendation. The authors should explain and highlight something really new to make it valuable in the scientific community if it is published.P

4.      To support the explanation in line 38-40, the authors are encouraged to adopt the suggested reference published by MDPI as follow: Computational Contact Pressure Prediction of CoCrMo, SS 316L and Ti6Al4V Femoral Head against UHMWPE Acetabular Cup under Gait Cycle. J. Funct. Biomater. 2022, 13, 64. https://doi.org/10.3390/jfb13020064

5.      The authors need to extend 1-2 sentences on the paragraph, it is should have 3 or more sentences to make it have enough explanation. For example in lines 105-108, and 109-112. Please revise it.

6.      For all of the figures, please arrange the figure with its description to fit on one page.

7.      Errors caused by surgeons need to be discussed further for diagnostic imaging strategy, such as ergonomic aspects.

8.      Line 662-672, polyethylene wear needs to explain related to materials selection since the present manuscript does not. My recommendation reference would be adopted.

9.      Line 681-689, the authors need to mention metal ions that are harmful to the reader. To support this statement, the previous reference is recommended to be adopted.

10.   discussions related to various failures due to operating procedures need to be disclosed.

11.   The limitation of the present review needs to be stated.

12.   Extending the conclusion is recommended since it does not summarize the whole of the content in the present review.

13.   Further research is recommended to include in the conclusion.

14.   Please recheck the English used in the present article.

Author Response

Dear Reviewer and Journal of Clinical Medicine editorial board,

Thank you for the opportunity of improving our manuscript (Round 1). It has been extensively modified and implemented with your recommendations.

A point-by-point rebuttal to the reviewer’s comments is provided below. All modifications to the text have been tracked.

I believe that the manuscript has been considerably improved by the modifications requested and I hope that you’ll find it suitable for publication in the Journal of Clinical Medicine in its present version.

A lot of new references have been added in this revised version. Please accept my apologize for bibliographic issues due to the revision on the downloaded file, that should be treated with Mrs Wu.

I remain at your disposal for any further requirements.

Sincerely,

Leading author

  1. Line 2-3, uppercase and lowercase of the title needs to be revised

Done, according to the template.

2.In the title of the present manuscript, it has “Part 2”? where is part 1, I have searched for it but have not found

The part 1 of this manuscript is about the pre-operative imaging work up and its recent updates. It is also under revision and almost accepted.

  1. What is the novelty of the present review? Imaging in total hip arthroplasty has been published in several previous works of literature. Based on the reviewer's evaluation the present article does not bring something new and serious scientific contribution which makes it not suitable for the recommendation. The authors should explain and highlight something really new to make it valuable in the scientific community if it is published.P

We totally agree with this comment. However, the litterature is increasing in this field. For the referring physician, this paper should be used as a guide for prescription when radiographs are inconclusive, depending on the clinical scenario, as an algorithm is proposed and radiographs are often insufficient when a prosthesis gets painful, except in case of evident complication. From the imaging point of view, the different imaging protocols are also discussed, from the “literature” point of view and with our clinical experience. The awaited appearance of all the complications and of normal aspects are explained and illustrated, for each arthroplasty type and with each imaging technique. This is therefore a helpful guide for all the physicians taking care of this kind of patients. Not all those aspects have already been described in one only paper to our knowledge, from this practical point of view.

  1. To support the explanation in line 38-40, the authors are encouraged to adopt the suggested reference published by MDPI as follow: Computational Contact Pressure Prediction of CoCrMo, SS 316L and Ti6Al4V Femoral Head against UHMWPE Acetabular Cup under Gait Cycle. J. Funct. Biomater. 2022, 13, 64. https://doi.org/10.3390/jfb13020064

We added this reference to the metal-on-metal section.

  1. The authors need to extend 1-2 sentences on the paragraph, it is should have 3 or more sentences to make it have enough explanation. For example in lines 105-108, and 109-112. Please revise it.

We have added explanations about the bearing surfaces as asked.

  1. For all of the figures, please arrange the figure with its description to fit on one page.

Done.

  1. Errors caused by surgeons need to be discussed further for diagnostic imaging strategy, such as ergonomic aspects.

We have added a section “ergonomics”, describing implants malpositioning and ischio-femoral impingement.

  1. Line 662-672, polyethylene wear needs to explain related to materials selection since the present manuscript does not. My recommendation reference would be adopted.

Done.

  1. Line 681-689, the authors need to mention metal ions that are harmful to the reader. To support this statement, the previous reference is recommended to be adopted.

Done.

  1. discussions related to various failures due to operating procedures need to be disclosed.

We have added a paragraph concerning periprosthetic fractures and dislocation depending on the surgical approaches.

  1. The limitation of the present review needs to be stated.

A limitation section has been added.

  1. Extending the conclusion is recommended since it does not summarize the whole of the content in the present review.

Done.

  1. Further research is recommended to include in the conclusion.

 Done.

  1. Please recheck the English used in the present article.

Done.

Reviewer 2 Report

  1. This manuscript summarized the application of radiographic assessment after hip arthroplasty. Until now, X-ray, CT scan, and MRI are normally applied in clinical medicine when dealing with postoperative complications. However, there are two questions to be answered before  publication of this manuscript.

    1.     This manuscript introduced the radiographic sign of several complications. Besides these signs, we care more about the prediction of these complications combining radiographic assessments, laboratory tests and other assessments. The author should add these diagnosis and prediction contents in this review.

    2.       This manuscript lacks introducing the relationship between radiographic sign and clinical symptom grading score including VAS and HHS. The author should add these contents in this review.

    3.       Considering the type of prosthesis and implants,there are several stem designs,based

    on the geometry of the prosthesis and fixation mechanism,which should be added into the manuscript

Author Response

Dear Reviewer and Journal of Clinical Medicine editorial board,

Thank you for the opportunity of improving our manuscript (Round 1). It has been extensively modified and implemented with your recommendations.

A point-by-point rebuttal to the reviewer’s comments is provided below. All modifications to the text have been tracked.

I believe that the manuscript has been considerably improved by the modifications requested and I hope that you’ll find it suitable for publication in the Journal of Clinical Medicine in its present version.

A lot of new references have been added in this revised version. Please accept my apologize for bibliographic issues due to the revision on the downloaded file, that should be treated with Mrs Wu.

I remain at your disposal for any further requirements.

Sincerely,

Leading author

  1. This manuscript summarized the application of radiographic assessment after hip arthroplasty. Until now, X-ray, CT scan, and MRI are normally applied in clinical medicine when dealing with postoperative complications. However, there are two questions to be answered before  publication of this manuscript.

  1. This manuscript introduced the radiographic sign of several complications. Besides these signs, we care more about the prediction of these complications combining radiographic assessments, laboratory tests and other assessments. The author should add these diagnosis and prediction contents in this review.

Clinical and biological tests concerning infection, impingement and adverse tissue reaction have been added in each section.

  1. This manuscript lacks introducing the relationship between radiographic sign and clinical symptom grading score including VAS and HHS. The author should add these contents in this review.

Those scores have been studied concerning post-operative analgesia, postoperative rehabilitation, supercapsular percutaneously assisted approach in total hip arthroplasty versus conventional, types of bearing surfaces and acetabular cup designs. VAS comparison about gluteal tendon repair and type of fixation have been added. To our knowledge, no correlation between those scales and a specific radiologic sign have been described.

  1. Considering the type of prosthesis and implants,there are several stem designs,based on the geometry of the prosthesis and fixation mechanism,which should be added into the manuscript

Those informations have been added and tracked.

Reviewer 3 Report

Interesting paper on postoperative imaging of THR. However, there many errors concerning hip implants and I would complete revision by an orthopaedic surgeon first before re-submission. English should also be revised.  

Here are a few comments (not exhaustive):

Page 1 Line 34: “about 27% of the patients complain of persistent post-surgical pain” this is the worse outcome reported in the literature, other studies find lower rates (for example: chronic pain following total hip arthroplasty: a nationwide questionnaire study. L Nikolajsen et al.Acta Anaesthesiol Scand. 2006 Apr;50(4):495-500. doi: 10.1111/j.1399-6576.2006.00976.x). Therefore, a range should be given.

Page 1 line 35-36: “follow-up surgery for about 1.3% of all total hip arthroplasty cases”: it is 1.3% per year, and it is all causes of complications together, not only due to pain. It can be infection, dislocation, loosening etc.

Page 1 line 36-37: complications depend also on patients’ comorbidities and activities.

Page 1 line 38 and 39: the sentence is not understood “undergoing HA” should be changed to “with a painful HA” or just “with HA”.

Page 2 line 43: “in case of complication” should be changed to “when a complication is suspected”

It should be explained that imaging in patients after HA is for:

-      Normal follow up, to detect complications with no clinical symptoms (polyethylene wear, evolutive osteolysis).

-      Diagnostic: to determine the cause or help to determine the cause of the patients symptoms

-      To guide for biopsy, fluid collection or other.

-      To plan for the treatment of the complication: in case of recurrent dislocation, version analysis of implants by CT-scan help to determine which one must be changed. In case of loosening of an implant, a CT scan is needed to determine the amount of bone loss around it in order to plan for the surgery (for example: amount of bone grafting needed, type of one grafting: auto, allograft, need for a metal augmentation of the acetabulum). Therefore, a CT-scan is not needed for the diagnosis of obvious loosening, but it is needed in case of loosening to determine the therapeutic strategy.

Figure 1b: the example should show the entire prosthesis, as it is important to analyze the entire implant-bone interface. Cemented prosthesis need a cement restrictor that must be seen on the X-ray.

-       Types of fixations: a phrase should be said about cement restrictor, because some of them have a metallic marker that can appear at the tip of the femoral stem. Also, many femoral stems now have a distal centralizer and its appearance should be explained.

Page 3 line 99: “or surfaces” should be changed to “or surface texture and coating”

Page 3 line 99: “and are pressed or screwed”: no, they are always impacted because they need a primary fixation, and screws can be added to secure the primary fixation. But screw alone is not possible.

Page 7 line 204: reference needed. Interest of EOS imaging.

Table 1: the values indicated need referencing.

Page 9, lines 227 and 228: revise English.

Figure 8a: the construct to measure FO is not right, the distance should be measured on the perpendicular line going from the center of the femoral head to the axis of the femoral stem. In the case shown, the FO line is not perpendicular to the stem axis.  

Page 12 line 307 and 308: “discrepancy was high (-1.4°)” this is confusing because achieving a precision of 1.4° is quite good, the 95% confidence interval should be given also.

Page 18: early dislocation is often due to implant mispositioning as opposed to late dislocations which is not due to implant mispositioning.

Chapter on complications: the imaging strategy is not properly explained.

Early dislocations: the main causes are muscle weakness, non-compliance with post-operative restrictions, traumatic and implant malpositioning. X-rays makes the diagnosis.

-        First episode: if trauma or an abnormal movement if found in the medical history prior to the dislocation, explanations are given to the patient to prevent some movements which are at risk of dislocation. If no trauma or abnormal movement is found prior to dislocation, post-operative muscular weakness is suspected. No need for further imaging.

-        Two or more episodes: CT-scan is necessary to determine implant version to make sure there is no malpositioning of the implants.

Late dislocation : the main causes are polyethylene wear, loosening of implant, muscle atrophy, trauma and non-compliance with post-operative restrictions. Implants malposition Is not probable as the patient had no dislocation in the early postoperative period. Implant malposition can be seen secondary to loosening: initially the implants were well positioned, then loosening made them move into wrong position, prone to dislocation. X-rays make the diagnosis and can find some of the causes: loosening of implants, polyethylene wear.  

-        First episode: if trauma or abnormal movement if found in the medical history prior to the dislocation and X-rays are normal, explanations are given to the patient to prevent some movements which are at risk of dislocation. If no abnormal movement is found prior to dislocation, Xrays are normal, muscular weakness is suspected. No need for further imaging.

-        two or more episodes: if X-rays are normal, muscular weakness is most probable. CT-scan is needed to search for loosening not seen on X-rays and to plan the revision surgery (for a dual mobility cup). MRI can be asked to search for Gluteus medius tendon tear, of fatty atrophy of gluteal muscles.

Page 19, line 456: Figure 15 “a peri-prosthetic dislocation” should be changed to “intra-prosthetic dislocation”.

Page 19, line 457: change word exenteration: ”…with bone kernel shows eccentric position of the femoral head…”

Page 21, line 514 and 515: reformulate phrase.

Better define loosening diagnosis which is implant mobilization on imaging techniques. This is the chapter where the authors have to insist on the need to visualize previous X-rays, even the postoperative X-rays on the best-case scenario.

Page 21, line 519: missing reference for the biology of particle wear biological reaction.

Figure 16, page 21 line 523: “mechanical and granulomatous loosening”: only the acetabular part is loose, not the stem.

Page 21, line 527: the varus position of the femoral stem is probably not due to loosening, and this is when the postoperative X-rays are necessary, to determine if the stem was already in this position.

Figure 18: all of the findings should be commented on the figure, such as the cerclage wire, the femorotomy etc.

Page 32, line 854-855: “concerning ceramic-on-ceramic prosthesis”: ceramic head can also rupture in case of ceramic on PE THR. “the femoral head fracture is secondary to trauma” reference needed.

Figure 21: d, e and f are images of a specific implant called sandwich ceramic liner, which are not used anymore because of a high rupture rate. There should be an explanation to why there is a radiolucent space between the metal back and the ceramic liner (polyethylene).

Page 32, line 874-875: reference needed.

Author Response

Dear Reviewer and Journal of Clinical Medicine editorial board,

Thank you for the opportunity of improving our manuscript (Round 1). It has been extensively modified and implemented with your recommendations.

A point-by-point rebuttal to the reviewer’s comments is provided below. All modifications to the text have been tracked.

I believe that the manuscript has been considerably improved by the modifications requested and I hope that you’ll find it suitable for publication in the Journal of Clinical Medicine in its present version.

A lot of new references have been added in this revised version. Please accept my apologize for bibliographic issues due to the revision on the downloaded file, that should be treated with Mrs Wu.

I remain at your disposal for any further requirements.

Sincerely,

Leading author

Interesting paper on postoperative imaging of THR. However, there many errors concerning hip implants and I would complete revision by an orthopaedic surgeon first before re-submission. English should also be revised.  

We have added precision about hip implants in the dedicated section, to precise femoral implants geometry.

English has also been revised.

Here are a few comments (not exhaustive):

Page 1 Line 34: “about 27% of the patients complain of persistent post-surgical pain” this is the worse outcome reported in the literature, other studies find lower rates (for example: chronic pain following total hip arthroplasty: a nationwide questionnaire study. L Nikolajsen et al.Acta Anaesthesiol Scand. 2006 Apr;50(4):495-500. doi: 10.1111/j.1399-6576.2006.00976.x). Therefore, a range should be given.

This reference has been added, along with “Beswick AD, Wylde V, Gooberman-Hill R, Blom A, Dieppe P. What proportion of patients report long-term pain after total hip or knee replacement for osteoarthritis? A systematic review of prospective studies in unselected patients. BMJ Open. 2012 Feb 22;2(1):e000435. doi: 10.1136/bmjopen-2011-000435. PMID: 22357571; PMCID: PMC3289991 ». pain is present in 7 to 27% of the patients and is of great concern in 12%.

Page 1 line 35-36: “follow-up surgery for about 1.3% of all total hip arthroplasty cases”: it is 1.3% per year, and it is all causes of complications together, not only due to pain. It can be infection, dislocation, loosening etc.

It has been precised.

Page 1 line 36-37: complications depend also on patients’ comorbidities and activities.

It has been precised.

Page 1 line 38 and 39: the sentence is not understood “undergoing HA” should be changed to “with a painful HA” or just “with HA”.

Done.

Page 2 line 43: “in case of complication” should be changed to “when a complication is suspected”

Done.

It should be explained that imaging in patients after HA is for:

-      Normal follow up, to detect complications with no clinical symptoms (polyethylene wear, evolutive osteolysis).

-      Diagnostic: to determine the cause or help to determine the cause of the patients symptoms

-      To guide for biopsy, fluid collection or other.

-      To plan for the treatment of the complication: in case of recurrent dislocation, version analysis of implants by CT-scan help to determine which one must be changed. In case of loosening of an implant, a CT scan is needed to determine the amount of bone loss around it in order to plan for the surgery (for example: amount of bone grafting needed, type of one grafting: auto, allograft, need for a metal augmentation of the acetabulum). Therefore, a CT-scan is not needed for the diagnosis of obvious loosening, but it is needed in case of loosening to determine the therapeutic strategy.

      This has been precised in the introduction. Those aspects are also developed in dedicated sections.

Figure 1b: the example should show the entire prosthesis, as it is important to analyze the entire implant-bone interface. Cemented prosthesis need a cement restrictor that must be seen on the X-ray.

This figure has been changed. A THA with a cement restrictor has been added (radiograph and CT).

-       Types of fixations: a phrase should be said about cement restrictor, because some of them have a metallic marker that can appear at the tip of the femoral stem. Also, many femoral stems now have a distal centralizer and its appearance should be explained.

      It has been added and a figure now shows a restrictor and a centralizer.

Page 3 line 99: “or surfaces” should be changed to “or surface texture and coating”

Done

Page 3 line 99: “and are pressed or screwed”: no, they are always impacted because they need a primary fixation, and screws can be added to secure the primary fixation. But screw alone is not possible.

Modified.

Page 7 line 204: reference needed. Interest of EOS imaging.

Done.

Table 1: the values indicated need referencing.

Done.

Page 9, lines 227 and 228: revise English.

Done.

Figure 8a: the construct to measure FO is not right, the distance should be measured on the perpendicular line going from the center of the femoral head to the axis of the femoral stem. In the case shown, the FO line is not perpendicular to the stem axis.  

It has been corrected.

Page 12 line 307 and 308: “discrepancy was high (-1.4°)” this is confusing because achieving a precision of 1.4° is quite good, the 95% confidence interval should be given also.

It has been clarified and completed.

Page 18: early dislocation is often due to implant mispositioning as opposed to late dislocations which is not due to implant mispositioning.

Chapter on complications: the imaging strategy is not properly explained.

Early dislocations: the main causes are muscle weakness, non-compliance with post-operative restrictions, traumatic and implant malpositioning. X-rays makes the diagnosis.

-        First episode: if trauma or an abnormal movement if found in the medical history prior to the dislocation, explanations are given to the patient to prevent some movements which are at risk of dislocation. If no trauma or abnormal movement is found prior to dislocation, post-operative muscular weakness is suspected. No need for further imaging.

-        Two or more episodes: CT-scan is necessary to determine implant version to make sure there is no malpositioning of the implants.

Late dislocation : the main causes are polyethylene wear, loosening of implant, muscle atrophy, trauma and non-compliance with post-operative restrictions. Implants malposition Is not probable as the patient had no dislocation in the early postoperative period. Implant malposition can be seen secondary to loosening: initially the implants were well positioned, then loosening made them move into wrong position, prone to dislocation. X-rays make the diagnosis and can find some of the causes: loosening of implants, polyethylene wear.  

-        First episode: if trauma or abnormal movement if found in the medical history prior to the dislocation and X-rays are normal, explanations are given to the patient to prevent some movements which are at risk of dislocation. If no abnormal movement is found prior to dislocation, Xrays are normal, muscular weakness is suspected. No need for further imaging.

-        two or more episodes: if X-rays are normal, muscular weakness is most probable. CT-scan is needed to search for loosening not seen on X-rays and to plan the revision surgery (for a dual mobility cup). MRI can be asked to search for Gluteus medius tendon tear, of fatty atrophy of gluteal muscles.

      We clarified this section according to the comments.

Page 19, line 456: Figure 15 “a peri-prosthetic dislocation” should be changed to “intra-prosthetic dislocation”.

Done.

Page 19, line 457: change word exenteration: ”…with bone kernel shows eccentric position of the femoral head…”

Done.

Page 21, line 514 and 515: reformulate phrase.

Better define loosening diagnosis which is implant mobilization on imaging techniques. This is the chapter where the authors have to insist on the need to visualize previous X-rays, even the postoperative X-rays on the best-case scenario.

Done.

Page 21, line 519: missing reference for the biology of particle wear biological reaction.

Those data have been added in the dedicated sections.

Figure 16, page 21 line 523: “mechanical and granulomatous loosening”: only the acetabular part is loose, not the stem.

It has been precised.

Page 21, line 527: the varus position of the femoral stem is probably not due to loosening, and this is when the postoperative X-rays are necessary, to determine if the stem was already in this position.

We did not have previous radiographs for this patient. The sentence has been removed to avoid any confusion.

Figure 18: all of the findings should be commented on the figure, such as the cerclage wire, the femorotomy etc.

Done.

Page 32, line 854-855: “concerning ceramic-on-ceramic prosthesis”: ceramic head can also rupture in case of ceramic on PE THR. “the femoral head fracture is secondary to trauma” reference needed.

Done.

Figure 21: d, e and f are images of a specific implant called sandwich ceramic liner, which are not used anymore because of a high rupture rate. There should be an explanation to why there is a radiolucent space between the metal back and the ceramic liner (polyethylene).

Done.

Page 32, line 874-875: reference needed.

Done.

Round 2

Reviewer 1 Report

I am recommending this manuscript for publication.